# K-Prism: A Knowledge-Guided and Prompt Integrated Universal Medical Image Segmentation Model

**Bangwei Guo[1], Yunhe Gao[2], Meng Ye[3], Difei Gu[1], Yang Zhou[1], Leon Axel[4], Dimitris Metaxas[1]**
[1]Rutgers University [2]Stanford University [3]The University of Texas at Arlington
[4]New York University
`bg654@rutgers.edu, dnm@cs.rutgers.edu`

## Abstract

Medical image segmentation is fundamental to clinical decision-making, yet existing models remain fragmented. They are usually trained on single knowledge sources and specific to individual tasks, modalities, or organs. This fragmentation contrasts sharply with clinical practice, where experts seamlessly integrate diverse knowledge: anatomical priors from training, exemplar-based reasoning from reference cases, and iterative refinement through real-time interaction. We present **K-Prism**, a unified segmentation framework that mirrors this clinical flexibility by systematically integrating three knowledge paradigms: (i) *semantic priors* learned from annotated datasets, (ii) *in-context knowledge* from few-shot reference examples, and (iii) *interactive feedback* from user inputs like clicks or scribbles. Our key insight is that these heterogeneous knowledge sources can be encoded into a dual-prompt representation: 1-D sparse prompts defining *what* to segment and 2-D dense prompts indicating *where* to attend, which are then dynamically routed through a Mixture-of-Experts (MoE) decoder. This design enables flexible switching between paradigms and joint training across diverse tasks without architectural modifications. Comprehensive experiments on 18 public datasets spanning diverse modalities (CT, MRI, X-ray, pathology, ultrasound, etc.) demonstrate that K-Prism achieves state-of-the-art performance across semantic, in-context, and interactive segmentation settings. Code is available at `https://github.com/bangwayne/K-Prism`.

## 1 Introduction

Medical image segmentation is a cornerstone of modern clinical workflows, supporting tasks such as tumor delineation (Heller et al., 2019; Bilic et al., 2023), organ quantification (Wasserthal et al., 2023), and vessel segmentation (Livne et al., 2019). While deep learning achieves strong results on individual benchmarks (Isensee et al., 2021; Hatamizadeh et al., 2022), real-world deployment remains challenging: healthcare institutions must maintain dozens of task-specific models tailored to different organs, modalities, and clinical scenarios, resulting in high deployment complexity and inconsistent performance (Zhou et al., 2021). This fragmentation stems from a deeper limitation: most existing models are constrained to a single knowledge type. They either depend on semantic priors learned from large labeled datasets (Liu et al., 2023a; Gao, 2024), adapt through in-context knowledge with few-shot examples (Butoi et al., 2023), or rely on interactive feedback (Ma et al., 2024; Isensee et al., 2025). Yet clinical practice is rarely confined to a single knowledge paradigm.

Consider a radiologist examining a rare pediatric tumor: they may leverage their semantic knowledge of anatomy, retrieve similar historical cases for reference, and iteratively refine boundaries through interactive feedback. Such flexible integration of diverse knowledge is routine for human experts, who dynamically adapt their strategies to the different clinical settings. In contrast, current AI models remain rigid, unable to seamlessly combine multiple knowledge sources. This forces clinicians to switch between separate models, disrupting workflows and limiting the potential of AI assistance. Recent efforts have attempted partial unification, combining two paradigms at a time (Gao et al., 2025; Guo et al., 2025; Wong et al., 2024), but no framework yet integrates all

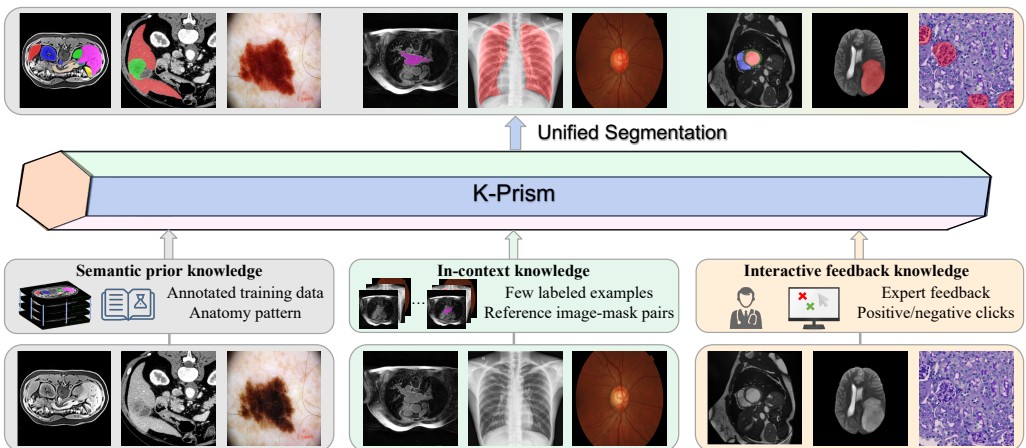

Figure 1: K-Prism integrates three forms of external knowledge, semantic priors (from annotated training datasets), in-context exemplars (from reference image–mask pairs), and interactive feedback (from user clicks and previous masks) into a single framework, enabling robust segmentation across diverse modalities and targets.

three knowledge types within a single architecture, see Table 1. This gap persists due to a key technical challenge: how to represent and process fundamentally different forms of knowledge within a unified framework that delivers strong performance across all modes.

We propose that achieving true universality in medical segmentation requires simultaneously addressing three clinically essential knowledge sources: (1) semantic prior knowledge from large-scale annotated datasets, capturing common anatomical and modality patterns, (2) in-context knowledge from reference examples, critical for rare diseases or new protocols where labeled data is scarce, (3) interactive feedback knowledge from user interactions such as clicks or scribbles, enabling iterative refinement. To realize this vision, we propose a **k**nowledge-guided and **pr**ompt-integrated universal medical **i**mage **s**egmentation **m**odel, **K-Prism**, which can integrate all three knowledge forms and adapt seamlessly across diverse inference scenarios and imaging domains (Figure 1).

To support joint training and inference across heterogeneous tasks and knowledge paradigms, we design a novel dual-prompt representation coupled with a Mixture-of-Experts (MoE) decoder. Our key insight is that diverse knowledge sources can be encoded into two complementary prompt types: 1-D sparse prompts to encode what to segment and 2-D dense prompts to encode where to attend, which are then dynamically routed through specialized experts based on the task requirements. This design enables K-Prism to not only match state-of-the-art (SOTA) performance in each individual paradigm but also support fluid transitions between modes, mirroring clinical workflows where different knowledge sources are combined based on availability and task demands. Beyond performance, K-Prism reduces deployment complexity by unifying all three paradigms, positioning itself as both a robust segmentation model and practical infrastructure for medical foundation models.

In summary, our key contributions are:

- We propose K-Prism, a unified framework and practical foundation for medical image segmentation that integrates three clinically relevant knowledge types.

- We design a dual-prompt representation that combines 1-D sparse and 2-D dense prompts with a MoE decoder for dynamic routing, enabling joint training across diverse tasks and modalities.

- We conduct comprehensive experiments on 18 datasets covering diverse anatomical targets, imaging modalities (CT, MRI, X-ray, pathology, etc.), and segmentation paradigms (semantic, in-context, interactive), achieving SOTA performance and strong cross-dataset generalization.

## 2 RELATED WORK

### 2.1 MEDICAL IMAGE SEGMENTATION UNDER DIVERSE KNOWLEDGE

Medical image segmentation has conventionally relied on task-specific, fully supervised models trained on well-annotated single-modality datasets (Isensee et al., 2021; Hatamizadeh et al., 2022). While effective in constrained settings, these models often fail to generalize to real-world scenarios where the available knowledge varies across cases (Zhou et al., 2022). To improve adaptability, recent studies have explored knowledge-driven strategies across diverse segmentation paradigms. In semantic segmentation, Liu et al. (2023a) leverage CLIP to encode anatomical descriptions into semantic priors, while Hermes (Gao, 2024) introduces learnable task-specific embeddings to guide a universal model. In in-context segmentation, Universeg (Butoi et al., 2023) and Tyche (Rakic et al., 2024) employ annotated support exemplars as visual references, enabling few-shot generalization across diverse tasks and datasets. In interactive segmentation, MedSAM (Ma et al., 2024) and nnInteractive (Isensee et al., 2025) refine model predictions by incorporating user prompts such as clicks and bounding boxes. Despite these advances, most existing methods are restricted to a single segmentation paradigm, limiting their ability to integrate heterogeneous knowledge sources.

### 2.2 UNIFIED SEGMENTATION FRAMEWORKS AND GENERALIZATION

Recent work has explored unified medical segmentation frameworks capable of integrating multiple tasks and knowledge types. As shown in Table 1, Iris (Gao et al., 2025) encodes reference image–label pairs into 1-D tokens, supporting both semantic and in-context segmentation. Verse (Guo et al., 2025) unifies semantic and interactive segmentation, enabling iterative refinement of initial predictions, while MultiverSeg (Wong et al., 2024) combines in-context and interactive segmentation to learn from a small number of annotated examples and improve results through expert feedback. However, no existing medical image segmentation framework integrates seman-

Table 1: Comparison of representative medical image segmentation methods across different paradigms.

| Method | Semantic | In-context | Interactive |
|---|:---:|:---:|:---:|
| nnU-Net (Isensee et al., 2021) | ✓ | – | – |
| UNETR (Hatamizadeh et al., 2022) | ✓ | – | – |
| Clip-driven (Liu et al., 2023a) | ✓ | – | – |
| Hermes (Gao, 2024) | ✓ | – | – |
| UniverSeg (Butoi et al., 2023) | – | ✓ | – |
| Tyche (Rakic et al., 2024) | – | ✓ | – |
| MedSAM (Ma et al., 2024) | – | – | ✓ |
| nnInteractive (Isensee et al., 2025) | – | – | ✓ |
| MultiverSeg (Wong et al., 2024) | – | ✓ | ✓ |
| Iris (Gao et al., 2025) | ✓ | ✓ | – |
| Verse (Guo et al., 2025) | ✓ | – | ✓ |
| **Ours (K-Prism)** | ✓ | ✓ | ✓ |

tic, in-context, and interactive feedback knowledge within a single architecture, leaving a gap for models that can adapt across all three paradigms.

## 3 METHOD

### 3.1 PROBLEM DEFINITION

Conventional medical image segmentation methods follow a knowledge-specific paradigm, where a dedicated model $f_{\theta_t}$ is trained for specific segmentation task $t$, relying solely on a single knowledge type such as semantic priors or in-context knowledge. K-Prism moves beyond this limitation by jointly integrating three complementary clinical knowledge sources, each aligned with a specific operational mode within a unified architecture:

- **Mode-1: Semantic segmentation** leverages learned class-level priors. Given a learnable embedding matrix $\boldsymbol{P} \in \mathbb{R}^{N_{\text{cls}} \times (p \times C)}$ and input image $\boldsymbol{I}$, where $N_{\text{cls}}$ is the number of classes in the training set, and each entry $\boldsymbol{p}_n \in \mathbb{R}^{p \times C}$ encodes semantic knowledge for class $n \in \{1, \ldots, N_{\text{cls}}\}$ through $p$ query vectors of dimension $C$, the model predicts $\hat{Y}_n = f_\theta(\boldsymbol{I} \mid \boldsymbol{p}_n)$.
- **Mode-2: In-context segmentation** uses reference examples to guide segmentation. Given a support set $\mathbb{S} = \{\boldsymbol{I}_{\text{ref}}, \boldsymbol{M}_{\text{ref}}\}$ containing reference images and masks, the model predicts $\hat{Y}_q = f_\theta(\boldsymbol{I}_q \mid \mathbb{S})$.

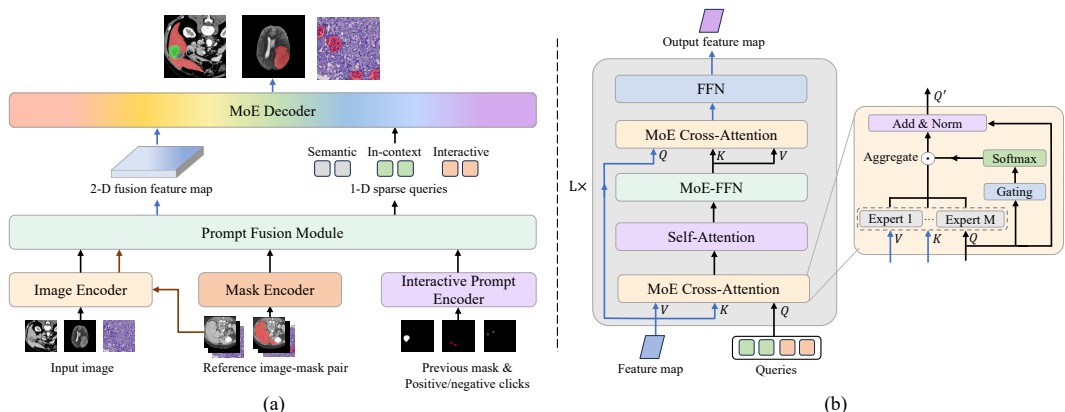

Figure 2: (a) Overview of the proposed K-Prism framework. Our model integrates three forms of external knowledge via the prompt fusion modules, encoding them into 1-D sparse queries and 2-D dense prompts to produce fusion feature maps. (b) The MoE decoder dynamically routes different prompts to specialized experts through cross-attention and gating, enabling task-aware specialization and robust segmentation across diverse scenarios.

- **Mode-3: Interactive segmentation** incorporates user feedback through clicks, enabling iterative refinement. Given click set $\mathbb{C} = \{c_i\}_{i=1}^{N_c}$, the model predicts $\hat{Y} = f_\theta(\boldsymbol{I} \mid \mathbb{C})$. This mode can also refine the initial predictions from Mode-1 and Mode-2.

In the following sections, we describe how K-Prism's unified architecture realizes each mode and seamlessly integrates multiple knowledge sources to achieve efficient, accurate, and broadly generalizable medical image segmentation.

## 3.2 INTEGRATING DIVERSE KNOWLEDGE INTO UNIFIED PROMPTS

Figure 2(a) shows the K-Prism framework. To process different knowledge types within a single architecture, we propose a dual-prompt design that captures two complementary aspects of segmentation by converting the inputs from each operational mode into a unified prompt representation: (i) 1-D sparse prompts, which define what the model should focus on by encoding task-level or instance-specific queries, and (ii) 2-D dense prompts, which indicate where the model should attend by modulating spatial feature maps to inject localization cues and refine structural details. Given an input image $\boldsymbol{I} \in \mathbb{R}^{3 \times H \times W}$, we first extract its feature map $\boldsymbol{F}$ through the encoder:

$$\boldsymbol{F} = \text{Encoder}(\boldsymbol{I}), \ \boldsymbol{F} \in \mathbb{R}^{C \times h \times w}, \tag{1}$$

where $C$ is the feature dimension and $h, w$ are the spatial resolutions of the encoded features.

### 3.2.1 SEMANTIC SEGMENTATION

In this mode, we use only 1-D sparse prompts, since semantic segmentation task relies solely on high-level class knowledge. The semantic prior embedding matrix $\boldsymbol{P} \in \mathbb{R}^{N_{\text{cls}} \times (p \times C)}$ is a learnable parameter, optimized through gradient backpropagation during training to learn class knowledge. To segment class $n$, the corresponding query set $\boldsymbol{p}_n$ is combined with the feature map $\boldsymbol{F}$ and fed into the decoder, without requiring any prompt fusion:

$$\text{Output}_n = \text{Decoder}(\boldsymbol{F} \mid \boldsymbol{p}_n). \tag{2}$$

### 3.2.2 IN-CONTEXT SEGMENTATION

This mode requires both 1-D sparse prompts and 2-D dense prompts. Let the support set be $\mathbb{S} = \{\boldsymbol{I}_{\text{ref}}, \boldsymbol{M}_{\text{ref}}\}$, where $\boldsymbol{I}_{\text{ref}} \in \mathbb{R}^{N_{\text{ref}} \times 3 \times H \times W}$ denotes the reference images and $\boldsymbol{M}_{\text{ref}} \in \mathbb{R}^{N_{\text{ref}} \times 1 \times H \times W}$ denotes the corresponding segmentation masks.

Given a query image $\boldsymbol{I}_q$ and the reference images $\boldsymbol{I}_{\text{ref}}$, the image encoder first extracts query and key features as shown in Figure 2(a):

$$\boldsymbol{F}_q = \text{Encoder}(\boldsymbol{I}_{\text{q}}), \boldsymbol{F}_k = \text{Encoder}(\boldsymbol{I}_{\text{ref}}), \tag{3}$$

and the reference image–mask pairs are encoded into value features via a lightweight mask encoder:

$$\boldsymbol{F}_v = \text{MaskEncoder}(Concat[\boldsymbol{I}_{\text{ref}}, \boldsymbol{M}_{\text{ref}}]). \tag{4}$$

For 2-D dense prompts, these encoded features are flattened and utilized in the prompt fusion:

$$\boldsymbol{K}^{\text{ref}} = \text{Flat}(\boldsymbol{F}_k) \in \mathbb{R}^{C \times N_{\text{ref}}hw}, \quad \boldsymbol{V}^{\text{ref}} = \text{Flat}(\boldsymbol{F}_v) \in \mathbb{R}^{C \times N_{\text{ref}}hw}, \quad \boldsymbol{K}^q = \text{Flat}(\boldsymbol{F}_q) \in \mathbb{R}^{C \times hw}. \tag{5}$$

In our design, $\boldsymbol{K}^{\text{ref}}$ and $\boldsymbol{K}^q$ are extracted by the same encoder, placing them in a shared feature space where the similarity matrix can be computed to align query and reference images. The value features $\boldsymbol{V}^{\text{ref}}$, derived from reference image–mask pairs, are then projected via this similarity matrix into the query-aligned space, transferring mask semantics to the query image representation. Both $\boldsymbol{K}^{\text{ref}}$ and $\boldsymbol{V}^{\text{ref}}$ thus serve as 2-D dense prompts. Formally, for any similarity function $c : \mathbb{R}^C \times \mathbb{R}^C \to \mathbb{R}$, we compute the softmax-normalized affinity matrix $\boldsymbol{W}$ by:

$$\boldsymbol{A}_{i,j} = c\big(\boldsymbol{K}^{\text{ref}}_{:,i}, \boldsymbol{K}^q_{:,j}\big), \quad \boldsymbol{W}_{i,j} = \frac{\exp(\boldsymbol{A}_{i,j})}{\sum_n \exp(\boldsymbol{A}_{n,j})}, \quad \boldsymbol{A}, \boldsymbol{W} \in \mathbb{R}^{N_{\text{ref}}hw \times hw}. \tag{6}$$

We use negative squared Euclidean distance for $c(\cdot, \cdot)$ following Cheng et al. (2021; 2024). The affinity matrix $\boldsymbol{W}$ projects the value features $\boldsymbol{V}^{\text{ref}}$ to align with the query features, producing the fusion feature map $\boldsymbol{F}_{\text{fuse}} = \boldsymbol{V}^{\text{ref}}\boldsymbol{W}$, $\boldsymbol{F}_{\text{fuse}} \in \mathbb{R}^{C \times hw}$, which is then passed to the decoder.

For the 1-D sparse prompt, given the encoded reference features $\boldsymbol{K}^{\text{ref}}$, we construct a set of $n_s$ object queries $\boldsymbol{Q}^s \in \mathbb{R}^{n_s \times C}$, where each query encodes compact, high-level object information. The first half represents the foreground, while the second half represents the background. A pooling mask $\boldsymbol{M}^{\text{pool}} \in [0, 1]^{N_{\text{ref}}hw}$ is obtained by downsampling and flattening $\boldsymbol{M}_{\text{ref}}$, and the $n$-th object query at location $i$ is then derived via masked average pooling:

$$\boldsymbol{m}_i^n = \begin{cases} 0, & n \leq \frac{n_s}{2} \text{ and } \boldsymbol{M}_i^{\text{pool}} < 0.5, \\ 0, & n > \frac{n_s}{2} \text{ and } \boldsymbol{M}_i^{\text{pool}} \geq 0.5, \\ 1, & \text{otherwise.} \end{cases} \quad \boldsymbol{Q}^s_{n,:} = \frac{\sum_{i=1}^{N_{\text{ref}}hw} (\boldsymbol{K}^{\text{ref}})_{i:}^{\top} \boldsymbol{m}_i^n}{\sum_{i=1}^{N_{\text{ref}}hw} \boldsymbol{m}_i^n}, \boldsymbol{Q}^s_{n,:} \in \mathbb{R}^C. \tag{7}$$

In summary, $\boldsymbol{Q}^s$ provides a 1-D sparse prompt encoding object-level foreground and background information, while $\boldsymbol{K}^{\text{ref}}$ and $\boldsymbol{V}^{\text{ref}}$ act as 2-D dense prompts for spatial modulation. This dual-prompt design enables the MoE decoder in the in-context setting to reason jointly over object semantics and spatial context. Further details and settings are provided in Appendix.

### 3.2.3 INTERACTIVE SEGMENTATION

For interactive segmentation, given an image $\boldsymbol{I}$ and user click set $\mathbb{C} = \{c_i\}_{i=1}^{N_c}$, we construct a three-channel prompt $\boldsymbol{I}_{\text{click}} \in \mathbb{R}^{3 \times H \times W}$ encoding positive clicks, negative clicks, and the previous mask prediction (Sofiiuk et al., 2022; Liu et al., 2023b). This prompt serves as a 2-D dense signal and is additively fused with the image features.

$$\boldsymbol{F}_{\text{fuse}} = \begin{cases} \text{PromptEncoder}(\boldsymbol{I}_{\text{click}}) + \text{Encoder}(\boldsymbol{I}), & \text{Mode-3 or refine Mode-1} \\ \text{PromptEncoder}(\boldsymbol{I}_{\text{click}}) + \boldsymbol{V}^{\text{ref}}\boldsymbol{W}, & \text{Refine Mode-2} \end{cases}, \quad \boldsymbol{F}_{\text{fuse}} \in \mathbb{R}^{C \times hw}. \tag{8}$$

Each click is also encoded as a 1-D sparse query. Following Guo et al. (2025), a click at image coordinates $(x, y)$ is mapped to feature map coordinates $(x', y') = (\lfloor x/s \rfloor, \lfloor y/s \rfloor)$, with $s$ denoting the downsampling ratio. Features within a local window of size $2r+1$ centered at $(x', y')$ are average-pooled, linearly transformed, and combined with a SAM-style positional embedding (Kirillov et al., 2023) to form the sparse query for that click. The $N_c$ clicks together form the query set $\boldsymbol{Q}^c \in \mathbb{R}^{N_c \times C}$, which is then fed into the decoder to produce the segmentation.

$$\text{Output} = \begin{cases} \text{Decoder}(\boldsymbol{F}_{\text{fuse}} \mid \boldsymbol{p}_n, \boldsymbol{Q}^c), & \text{Refine Mode-1} \\ \text{Decoder}(\boldsymbol{F}_{\text{fuse}} \mid \boldsymbol{Q}^s, \boldsymbol{Q}^c), & \text{Refine Mode-2} \\ \text{Decoder}(\boldsymbol{F}_{\text{fuse}} \mid \boldsymbol{Q}^c), & \text{Mode-3} \end{cases} \tag{9}$$

## 3.3 MIXTURE-OF-EXPERTS CROSS-ATTENTION DECODER

The unified nature of K-Prism makes it challenging for a single decoder to effectively process fundamentally different knowledge types, as each requires distinct processing strategies. A standard decoder would struggle to optimize across all three paradigms simultaneously, potentially leading to suboptimal performance. To address this, we introduce a Mixture-of-Experts (MoE) decoder that enables dynamic, task-aware specialization while maintaining shared representational capacity. Our decoder adopts a bidirectional cross-attention design to effectively fuse 1-D sparse prompt and 2-D fusion feature maps (Figure 2(b)). In the first layer, 1-D sparse prompts are projected into queries $\boldsymbol{Q} \in \mathbb{R}^{q \times C}$, while 2-D fusion feature maps are projected into keys and values $\boldsymbol{K}, \boldsymbol{V} \in \mathbb{R}^{hw \times C}$. MoE is applied to both cross-attention (CA) and the feed-forward network (FFN). And each MoE-CA layer consists of $M$ multi-head attention experts $\{\mathcal{A}_m\}_{m=1}^M$, where each computes the cross-attention: $\boldsymbol{O}_m = \mathcal{A}_m(\boldsymbol{Q} + \boldsymbol{P}_Q, \boldsymbol{K} + \boldsymbol{P}_K, \boldsymbol{V})$, $\boldsymbol{O}_m \in \mathbb{R}^{q \times C}$, with $\boldsymbol{P}_Q$ and $\boldsymbol{P}_K$ denoting positional embeddings. Cross-attention is given by:

$$\text{CrossAttn}_m(\boldsymbol{Q}, \boldsymbol{K}, \boldsymbol{V}) = \text{softmax}\left(\frac{\boldsymbol{Q}\boldsymbol{K}^\top}{\sqrt{C}}\right)\boldsymbol{V}, \tag{10}$$

To adaptively combine the outputs of multiple experts, a gating network $G : \mathbb{R}^{q \times C} \to \mathbb{R}^{q \times M}$ predicts query-specific expert weights: $\boldsymbol{\alpha} = \text{softmax}(G(\boldsymbol{Q}))$, $\boldsymbol{\alpha} \in \mathbb{R}^{q \times M}$. The expert outputs are stacked as $\boldsymbol{O}^{\text{stack}} \in \mathbb{R}^{q \times M \times C}$, and the gating weights are broadcast to $\boldsymbol{\alpha}^b \in \mathbb{R}^{q \times M \times 1}$. The final MoE output is then obtained via element-wise multiplication and summation:

$$\boldsymbol{O}_{\text{moe}} = \sum_{m=1}^M \boldsymbol{\alpha}^b_{:,m,:} \odot \boldsymbol{O}^{\text{stack}}_{:,m,:}, \quad \boldsymbol{O}_{\text{moe}} \in \mathbb{R}^{q \times C}. \tag{11}$$

We then apply residual addition and layer normalization to obtain the updated query representation $\boldsymbol{Q}'$, which is then passed to Self-Attention and MoE-FFN blocks to further update. In the second MoE cross-attention layer, the updated 1-D queries are projected into keys and values, while the 2-D fusion feature maps serve as queries, establishing bidirectional interaction between the sparse queries and fusion feature maps. More details of the decoder are provided in Appendix.

# 4 EXPERIMENT

## 4.1 EXPERIMENTAL SETUP

**Datasets.** We train on 12 publicly available datasets (Ji et al., 2022; Campello et al., 2021; Bilic et al., 2023; Heller et al., 2019; Li et al., 2020; 2021; 2022a;b; Al-Dhabyani et al., 2020; Jaeger et al., 2013; Candemir et al., 2013; Deng et al., 2025; Kovalyk et al., 2022; Ngoc Lan et al., 2021; Codella et al., 2019; Tschandl et al., 2018) spanning diverse imaging modalities (CT, MRI, pathology, ultrasound, etc.) and clinical targets (organs, tumors, lesions, etc.). For evaluation, we use (i) four external datasets: BTCV (Landman et al., 2015), ACDC (Bernard et al., 2018), UW-SC (University of Waterloo), and BUS (Yap et al., 2020), to test cross-dataset generalization, and (ii) two unseen-class datasets: BraTS (Baid et al., 2021) and M&Ms-2 (Campello et al., 2021), to assess adaptation to novel structures. Details are provided in Appendix.

**Baselines.** We compare against SOTA methods in three categories: (1) Semantic segmentation: the task-specific model nnU-Net (Isensee et al., 2021), and universal models Clip-driven (Liu et al., 2023a), UniSeg (Ye et al., 2023), and Hermes (Gao, 2024); (2) In-context segmentation: UniverSeg (Butoi et al., 2023), Tyche (Rakic et al., 2024), MultiverSeg (Wong et al., 2024), and Iris (Gao et al., 2025); (3) Interactive segmentation: nnInteractive (Isensee et al., 2025), MedSAM (Ma et al.,

Table 2: Comparison of semantic and in-context segmentation across different in-distribution datasets, measured by mean Dice scores (%). All in-context models use one-shot inference.

| Method | AMOS CT | AMOS MRI | M&Ms | LiTS Tumor | KiTS Tumor | LAScarQS | Breast Cancer | Chest X-ray | KPIs | PAPILA | BKAI POLY | ISIC | AVG |
|---|---|---|---|---|---|---|---|---|---|---|---|---|---|
| *Semantic (Task-specific Model)* | | | | | | | | | | | | | |
| nnU-Net | 76.26 | 76.57 | 85.42 | 56.84 | 73.39 | **84.28** | 71.68 | 94.95 | 86.22 | **96.17** | 83.45 | 86.76 | 81.00 |
| *Semantic (Universal Models)* | | | | | | | | | | | | | |
| Clip-driven | 84.37 | 83.20 | 84.10 | 60.12 | 75.99 | 80.94 | 76.47 | 95.45 | 85.82 | 95.81 | 86.32 | 86.04 | 84.31 |
| UniSeg | 84.91 | **85.36** | 85.34 | 56.97 | 73.41 | 81.57 | 75.71 | **96.02** | 80.88 | 91.40 | 82.57 | 84.24 | 83.96 |
| Hermes | 85.06 | 84.65 | 85.87 | 58.66 | 73.64 | 83.30 | **78.09** | 95.64 | 85.41 | 95.87 | 85.59 | 88.64 | 85.02 |
| K-Prism | **85.21** | 84.39 | **86.15** | **64.22** | **78.70** | 83.27 | 76.86 | 95.93 | **86.68** | 95.71 | **86.76** | 89.36 | **86.21** |
| *In-context* | | | | | | | | | | | | | |
| UniverSeg | 60.51 | 59.02 | 60.51 | 49.02 | 58.60 | 72.18 | 54.82 | 94.65 | 81.07 | 95.45 | 73.93 | 80.01 | 71.53 |
| Tyche | 61.62 | 59.52 | 61.21 | 50.01 | 59.68 | 72.26 | 54.94 | 94.42 | 81.05 | 95.60 | 74.73 | 80.80 | 72.12 |
| MultiverSeg | 58.48 | 58.13 | 58.83 | 50.60 | 65.43 | 67.74 | 67.02 | 94.01 | 82.36 | 94.98 | 72.77 | 85.10 | 72.41 |
| Iris | 74.10 | 76.59 | 83.37 | 59.26 | 74.65 | 78.14 | 75.47 | **95.86** | 83.96 | 95.31 | 82.96 | 85.87 | 81.76 |
| K-Prism | **79.72** | **78.21** | **85.22** | **62.93** | **79.12** | **81.22** | 75.55 | 95.80 | **86.48** | 95.74 | **84.88** | **89.78** | **84.82** |

Table 3: Comparison of semantic and in-context segmentation across external and unseen-class datasets, measured by mean Dice scores (%). All in-context models use one-shot inference.

| Method | External | | | | | Unseen-Class | | |
|---|---|---|---|---|---|---|---|---|
| | BTCV | ACDC | UW-SC | BUS | AVG | BraTS | M&Ms-2 | AVG |
| *Semantic (Task-specific Model)* | | | | | | | | |
| nnU-Net | 67.25 | 84.64 | 79.78 | 68.00 | 74.92 | - | - | - |
| *Semantic (Universal Models)* | | | | | | | | |
| Clip-driven | 78.43 | 85.35 | 83.19 | 69.86 | 79.21 | - | - | - |
| UniSeg | **80.89** | 86.69 | 84.47 | 69.24 | 80.82 | - | - | - |
| Hermes | 79.25 | 85.87 | **87.64** | 72.49 | 81.81 | - | - | - |
| K-Prism | 80.24 | **87.26** | 87.54 | **78.75** | **83.45** | - | - | - |
| *In-context* | | | | | | | | |
| UniverSeg | 55.63 | 56.87 | 75.08 | 47.56 | 58.29 | 15.61 | 21.16 | 18.39 |
| Tyche | 57.07 | 57.62 | 74.60 | 43.68 | 58.74 | 15.92 | 21.95 | 18.94 |
| MultiverSeg | 48.69 | 51.15 | 78.76 | 62.08 | 60.67 | 18.23 | 26.21 | 22.22 |
| Iris | 74.97 | 83.36 | 86.82 | 68.93 | 78.52 | **25.83** | 26.30 | 26.07 |
| K-Prism | **76.82** | **85.89** | **87.66** | **79.59** | **82.49** | 22.20 | **41.61** | **31.91** |

2024), MultiverSeg (Wong et al., 2024), SAM2 (Ravi et al., 2024), and SegNext (Liu et al., 2024a). All models are trained with 2-D slices extracted from our curated datasets under identical conditions. Implementation details are described in the Appendix.

## 4.2 RESULTS

### 4.2.1 SEMANTIC SEGMENTATION

Table 2 shows that K-Prism achieves the highest average Dice scores (86.21%) across 12 in-distribution datasets of diverse modalities, surpassing Hermes (85.02%), Clip-driven (84.31%), and UniSeg (83.96%). As a task-specific model, nnU-Net performs well on certain single-organ datasets (e.g., PAPILA 96.17%, LAScarQS 84.28%), consistent with its per-dataset optimization strategy (Isensee et al., 2021). However, its accuracy drops notably on multi-organ 2D slice segmentation (Liu et al., 2024b), such as AMOS-MRI (76.57%), and declines further on external datasets. In contrast, universal models achieve more stable performance across modalities, and K-Prism continues this trend with even greater consistency. On challenging tasks such as tumor segmentation, it achieves notable performance improvements: 64.22% for LiTS and 78.70% for KiTS. These results demonstrate the effectiveness of our framework in capturing anatomical context and delivering robust segmentation. Table 3 presents results on external datasets. Our model yields the best generalization across datasets with 83.45% average Dice score. These results establish K-Prism as the new state-of-the-art for semantic medical segmentation, with strong generalization across diverse modalities and clinical settings.

Table 4: Comparison of interactive segmentation performance across in-distribution, external and unseen-class datasets. To maintain clarity, only the mean values of key metrics are reported across multiple datasets. NoC90/NoC95 denote the average number of clicks required to reach 90% and 95% Dice scores, while Dice(1)/Dice(5) refer to Dice scores after 1 and 5 clicks, respectively.

| Method | In-distribution | | | | External | | | | Unseen-Class | | | |
|---|---|---|---|---|---|---|---|---|---|---|---|---|
| | NoC90↓ | NoC95↓ | Dice(1)↑ | Dice(5)↑ | NoC90↓ | NoC95↓ | Dice(1)↑ | Dice(5)↑ | NoC90↓ | NoC95↓ | Dice(1)↑ | Dice(5)↑ |
| *Interactive* | | | | | | | | | | | | |
| nnInteractive | 3.44 | 5.08 | 73.36 | 88.44 | 3.52 | 5.58 | 68.43 | 85.53 | 4.96 | 7.42 | 47.60 | 86.77 |
| MedSAM | 3.17 | 5.16 | 88.23 | 91.21 | 3.35 | 6.40 | 87.26 | 90.93 | 6.69 | 8.88 | 65.44 | 80.13 |
| MultiverSeg | 3.32 | 5.19 | 70.57 | 92.97 | 3.27 | 5.64 | 73.45 | 92.80 | 5.47 | 7.98 | 53.15 | 87.93 |
| SAM2 | 3.94 | 6.27 | 86.46 | 87.96 | 3.59 | 7.34 | 87.57 | 89.00 | 8.71 | 9.84 | 59.88 | 66.81 |
| SegNext | 2.50 | 4.08 | 89.53 | 93.80 | 2.63 | 5.00 | 88.43 | 92.96 | 4.77 | 7.22 | **71.99** | 87.72 |
| K-Prism | **1.95** | **3.51** | **89.55** | **95.50** | **2.01** | **4.24** | **88.67** | **94.92** | **4.32** | **6.62** | 68.67 | **90.67** |

### 4.2.2 IN-CONTEXT SEGMENTATION

Table 2 (bottom) summarizes in-context segmentation performance across 12 in-distribution datasets. Our method achieves the highest average Dice score (84.82%) and ranks first on 11 out of 12 datasets. Table 3 further evaluates generalization to external and unseen-class datasets. Across four external datasets (BTCV, ACDC, UW-SC, and BUS), our method consistently outperforms all prior in-context segmentation models, achieving an average Dice score of 82.49%. In the unseen-class setting, our model attains 31.91% on average, including a notable 15% improvement over Iris on M&Ms-2 (41.61% vs. 26.30%), demonstrating strong adaptability to novel anatomical structures under limited supervision.

Compared to semantic segmentation, the in-context segmentation mode of K-Prism shows highly competitive performance—even surpassing Mode-1 (semantic segmentation) on challenging datasets such as KiTS (79.12% vs. 78.70%) and ISIC (89.78% vs. 89.36%), though performance drops are observed on AMOS_CT (85.21% → 79.72%), AMOS_MRI (84.39% → 78.21%) and BTCV (80.24% → 76.82%), due to the complexity of multi-organ segmentation tasks where 2-D reference slices provide limited anatomical context. Despite this, K-Prism sets a new SOTA for in-context medical segmentation and remains competitive with fully supervised counterparts, highlighting its effectiveness and potential for future in-context learning frameworks.

### 4.2.3 INTERACTIVE SEGMENTATION

We further evaluate our model in the interactive segmentation setting, where user clicks iteratively refine predictions (Table 4). Our method consistently achieves the best performance across in-distribution, external, and unseen-class datasets. With five clicks, our model reaches a 95.50% Dice score on in-distribution datasets, surpassing strong baselines like SegNext (93.80%), SAM2 (87.96%), MedSAM (91.21%) and MultiverSeg (92.97%). Notably, our method also achieves the lowest NoC90 and NoC95 (1.95 and 3.51, respectively), indicating high-accuracy predictions with fewer interactions. On external datasets, our model maintains strong generalization, attaining a 94.92% Dice score at five clicks, along with the best NoC scores (2.01 and 4.24). For unseen-class datasets, our method delivers clear advantages, reaching a 90.67% Dice score at five clicks, while maintaining efficient convergence (4.32 for NoC90).

To further examine model behavior during interactive segmentation, we analyze the convergence curves (Figure 3). CNN-based methods, nnInteractive and MultiverSeg, that emphasize local structural biases, show low initial Dice scores but improve rapidly with more clicks. SAM-family models, SAM2 and MedSAM, achieve relatively high initial Dice scores but show only marginal gains with additional clicks. Because clicks are encoded only as 1-D sparse points without any 2-D spatial point map, these models show very limited improvement even with more user feedback. SegNext achieves a better balance with strong initialization and steady improvement. In comparison, our model which fuses 1-D sparse and 2-D dense prompts, achieves the best of both: high starting Dice and click-efficiency, demonstrating the effectiveness of unified prompt integration for precise and efficient interaction.

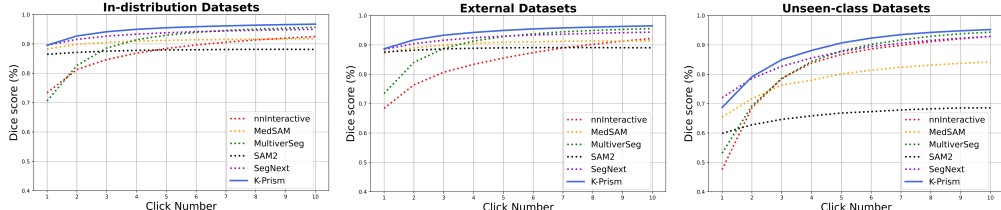

Figure 3: Convergence curves of interactive segmentation on in-distribution, external, and unseen-class datasets. K-Prism consistently achieves higher Dice scores and faster convergence compared to all baselines.

Table 5: K-Prism ablations.

(a) MoE-based Cross-Attention (CA) and FFN layers contribute to performance gains. We remove one component at a time.

(b) Evaluates the impact of different components in the in-context segmentation mode.

| Method | # Params | Semantic Dice | In-context Dice | Interactive NoC90↓ | NoC95↓ | Dice(1)↑ | Dice(5)↑ |
|---|---|---|---|---|---|---|---|
| Ours | 43.29M | 81.28 | 79.21 | 2.31 | 4.80 | 86.76 | 93.79 |
| w/o MoE CA | 30.64M | 77.38 | 77.11 | 2.47 | 5.11 | 86.55 | 93.23 |
| w/o MoE FFN | 40.13M | 78.57 | 78.37 | 2.37 | 4.84 | 86.50 | 93.67 |
| w/o MoE FFN & CA | 27.48M | 76.77 | 75.10 | 2.62 | 5.04 | 84.16 | 92.22 |

| Method | Dice |
|---|---|
| Ours | 80.84 |
| w/o 2-D fusion | 54.65 |
| w/o 1-D queries | 77.19 |

(c) Evaluates the impact of different components in the interactive segmentation mode.

(d) Impact of refinement in interactive segmentation on the ACDC and BUS dataset.

| Method | NoC90↓ | NoC95↓ | Dice(1)↑ | Dice(5)↑ |
|---|---|---|---|---|
| Ours | 2.34 | 4.97 | 86.55 | 93.68 |
| w/o 2-D fusion | 5.07 | 7.84 | 77.65 | 79.21 |
| w/o 1-D queries | 2.57 | 5.01 | 85.44 | 93.39 |

| Method | ACDC NoC90↓ | NoC95↓ | BUS NoC90↓ | NoC95↓ |
|---|---|---|---|---|
| Mode-3 | 2.67 | 4.97 | 1.67 | 4.05 |
| Mode-1 refine | 1.67 | 4.52 | 1.10 | 3.47 |
| Mode-2 refine | 1.77 | 4.59 | 1.17 | 3.53 |

## 4.3 Ablation

This section covers component ablations (Tables 5(a)–(c)) and a separate refinement-efficiency study (Table 5(d)). Parts (a)–(c) are based on models trained on M&Ms and Breast Cancer and evaluated on ACDC and BUS. Part (a) examines the impact of MoE components across all modes: removing either MoE cross-attention or the MoE FFN noticeably degrades performance, while removing both leads to the largest drop. Parts (b) and (c) evaluate single-mode settings (in-context and interactive, respectively). In both cases, 2-D fusion proves indispensable—its removal nearly collapses performance—whereas removing 1-D queries causes only a modest decline, showing that they are complementary but less critical.

Table 5(d) evaluates refinement efficiency in the interactive setting using the model trained in our main experiment. Mode-3 serves as a pure interactive baseline, while Mode-1 and Mode-2 leverage semantic or in-context knowledge to provide an initial mask, which is then refined with user clicks. With such initialization, the required clicks to reach 90% Dice scores decrease by over 30%. This initialization strategy transforms the clinical workflow from manual delineation to efficient refinement, substantially reducing annotation burden while maintaining flexibility through seamless mode switching. Further ablation studies and analyses are included in Appendix.

## 5 Discussion

Our work presents a unified framework for medical image segmentation that integrates semantic priors, in-context, and interactive feedback knowledge. While the results demonstrate strong performance and generalization, several aspects merit further discussion.

K-Prism provides a flexible framework that unifies different knowledge sources, enabling seamless switching across paradigms. This positions it as a foundation for universal medical segmentation.

Future work may refine components like semantic priors, in-context retrieval, or interactive refinement, making K-Prism not just a model but a platform for ongoing innovation. Our model can also serve as an efficient annotation tool for building large-scale medical image datasets. Clinicians can generate initial segmentation from semantic priors or in-context inference and refine them with minimal clicks, creating a human-in-the-loop pipeline that highlights K-Prism's value as both a segmentation framework and a practical infrastructure for large-scale medical modeling.

Despite these strengths, several challenges remain. The accuracy of in-context segmentation still relies on the quality of reference examples, and the MoE decoder introduces additional computational cost that may hinder real-time deployment. Furthermore, large domain shifts, such as unseen-class datasets or new imaging protocols, continue to pose difficulties, often leading to degraded performance. In unseen-class scenarios, we observe that K-Prism's interactive mode remains highly robust, as click-based refinement relies on local boundary and texture cues rather than global semantic priors, enabling reliable adaptation even to anatomy never seen during training. In contrast, in-context segmentation exhibits larger performance degradation under severe anatomical or modality shifts, reflecting the intrinsic difficulty of exemplar-based matching when the reference and query differ substantially. These observations highlight both the strengths and natural limitations of different knowledge paradigms, and they further demonstrate the strength of our unified design, which maintains reliable performance by allowing interactive refinement to compensate when exemplar-guided inference becomes unreliable. Furthermore, our current 2D slice-based formulation inherently limits the full utilization of 3D volumetric context. Although 2D interactive refinement aligns well with real clinical workflows, where radiologists typically correct boundaries on a slice-by-slice basis, future work could explore efficient 2D-to-3D propagation strategies that transfer refined slice-level predictions to full volumetric outputs. Additional research may also focus on (i) optimizing expert specialization across segmentation paradigms and (ii) enhancing robustness for unseen anatomical classes and domain-shifted scenarios.

## 6 CONCLUSION

We introduced K-Prism, a unified segmentation framework that integrates semantic priors, in-context, and interactive feedback knowledge into a unified dual-prompt representation, coupled with a Mixture-of-Experts decoder for dynamic routing. This design supports flexible adaptation across clinical scenarios, from supervised segmentation to few-shot adaptation and interactive refinement. Experiments on 18 public datasets show consistent state-of-the-art performance with strong robustness and transferability to unseen domains. We view this work as a step toward universal medical image segmentation models that can serve as reliable backbones for diverse clinical applications, narrowing the gap between algorithmic advances and real-world deployment.

## 7 ACKNOWLEDGMENTS

This research has been partially funded by research grants to D. Metaxas through NSF: 2310966, 2235405, 2212301, 2003874, 1951890 and NIH 2R01HL127661.

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

# A APPENDIX

## A.1 DATASET

### A.1.1 DATASET DETAILS

**Abdominal Multi-Organ Segmentation (AMOS) (Ji et al., 2022).** The AMOS dataset comprises 500 CT and 100 MRI abdominal scans from 600 patients at Longgang District People's Hospital. It provides annotations for 13 anatomical structures, including spleen, right kidney, left kidney, gallbladder, esophagus, liver, stomach, aorta, inferior vena cava, pancreas, right adrenal gland, left adrenal gland, duodenum, bladder, and prostate/uterus. The AMOS_CT set includes 200 training and 100 validation cases, while the AMOS_MRI set offers 40 training and 20 validation cases. We use both modalities for upstream training with a 95%/5% split on the official training set to train and validate, then test on the provided test set.

**Multi-Centre, Multi-Vendor & Multi-Disease Cardiac Image Segmentation Challenge (M&Ms) (Campello et al., 2021).** The M&Ms dataset from the MICCAI 2020 Challenge includes multi-center, multi-vendor cardiac MRI scans from patients with cardiomyopathies and healthy controls. All scans are short-axis cine images and have expert annotations for left/right ventricles and left ventricular myocardium at end-diastolic and end-systolic phases. The official train and test sets are used.

**The Liver Tumor Segmentation Benchmark (LiTS) (Bilic et al., 2023).** This dataset includes 201 contrast-enhanced abdominal CT scans (131 training, 70 testing) from multiple international medical centers. It covers patients with diverse liver tumors, including hepatocellular carcinoma and metastases from colorectal, breast, and lung cancers. All scans contain expert annotations for liver and tumor regions. We use the 131 public training cases, split 80%/20%, for upstream training and testing. In our experiments, only the tumor labels from each CT volume are utilized.

**The Kidney Tumor Segmentation 2019 dataset (KiTS) (Heller et al., 2019),** This dataset is collected at the University of Minnesota Medical Center between 2010 and 2018, includes CT scans and clinical treatment outcomes from 300 patients who underwent nephrectomy for kidney tumors. Of these, 210 cases are publicly available, while the remaining 90 are reserved for evaluation. In our upstream training, we utilize the public portion by splitting it into 80% for training, and 20% for testing. In our experiments, only the tumor labels from each CT volume are utilized.

**Left Atrial and Scar Quantification & Segmentation Challenge (LAScarQS) (Li et al., 2020; 2021; 2022a;b).** The LAScarQS dataset was released as part of a MICCAI challenge dedicated to left atrial (LA) cavity and scar segmentation from late gadolinium enhancement (LGE) cardiac MRI. It comprises 130 LGE MRI volumes from patients with atrial fibrillation (AF), acquired across multiple clinical centers. Each volume contains expert annotations for both the LA cavity and atrial scar regions. The dataset reflects substantial real-world variability in image quality, atrial morphology, and scar patterns, with many cases presenting significant segmentation challenges. Following the official protocol, we adopt the 80%/20% split of the training set for model training and testing. In our experiments, only the LA cavity annotations are used.

**Dataset of Breast Ultrasound Images (Breast Cancer) (Al-Dhabyani et al., 2020).** This dataset comprises 780 grayscale ultrasound images collected from 600 female patients (aged 25–75) at Baheya Hospital for Early Detection and Treatment of Women's Cancer, Egypt. Each image is categorized into one of three classes: normal, benign, or malignant. Expert-annotated segmentation masks are provided for all lesion-containing images. The dataset reflects real-world variability in breast anatomy, lesion characteristics, and image quality, making it valuable for developing and evaluating models for breast cancer classification, detection, and segmentation. We use only the 647 images labeled as benign or malignant and adopt an 80%/20% split for training and testing.

**Chest X-ray Masks and Labels (Chest X-ray) (Al-Dhabyani et al., 2020; Jaeger et al., 2013).** Chest X-ray is a chest radiograph dataset provided by the National Library of Medicine, National Institutes of Health (Bethesda, USA) and Shenzhen No.3 People's Hospital (Guangdong, China). This dataset includes a wide spectrum of abnormalities such as effusions and miliary patterns and is widely used for tuberculosis screening, lung segmentation, and domain adaptation studies. In this study, we only use the lung segmentation mask to train our models. We adopt an 80%/20% split of the training set for model training and testing.

**Kidney Pathology Image Segmentation (KPIs) Challenge (Deng et al., 2025).** The KPIs dataset was released as part of the MICCAI 2024 Challenge to benchmark glomeruli segmentation performance in chronic kidney disease (CKD) pathology. We focus on the patch-level segmentation task, which involves pixel-wise identification of glomeruli within PAS-stained image patches. The patches exhibit variations in glomeruli morphology and surrounding tissue structures due to differences in disease states and slide preparation. We adopt a 75%/5%/20% split on the official training set for training, validation, and testing. The original images are then uniformly partitioned into non-overlapping patches of size $512 \times 512$, and patches without any glomeruli annotations are discarded.

**PAPILA Dataset (Kovalyk et al., 2022).** The PAPILA dataset provides fundus photographs from both eyes of individual patients, accompanied by expert annotations. Each image is annotated with optic disc and optic cup segmentations, while patient-level clinical labels are available for disease evaluation. In this study, we focus exclusively on the optic disc segmentation task, using the provided disc masks to train and evaluate our models. We adopt a 80%/20% split on the official dataset with 488 image-mask pairs for training and testing.

**BKAI-IGH NeoPolyp Dataset (BKAI_POLY) (Ngoc Lan et al., 2021).** The BKAI-IGH NeoPolyp dataset, released by the BKAI Research Center (Hanoi University of Science and Technology) in collaboration with the Institute of Gastroenterology and Hepatology (IGH), Vietnam, consists of 1,200 colonoscopy images (1,000 white-light imaging (WLI) and 200 flexible spectral imaging color enhancement (FICE) images). The dataset is split into 1,000 training and 200 test images. Each polyp is annotated with both segmentation masks and binary labels indicating neoplastic (red) or non-neoplastic (green) classes, verified by two experienced endoscopists. We adopt an 80%/20% split on the official training set for training and testing.

**International Skin Imaging Collaboration (ISIC) Dataset (Tschandl et al., 2018; Codella et al., 2019).** The ISIC dataset is a large-scale dermoscopic image collection, introduced through the ISIC Skin Lesion Analysis Challenges. It contains high-resolution skin lesion images accompanied by expert-annotated segmentation masks delineating lesion boundaries. In this study, we use the ISIC 2018 segmentation subset. The dataset encompasses diverse lesion appearances and acquisition conditions, capturing real-world variability in skin tone, lighting, and lesion morphology. We adopt the official dataset split, using it for model training, validation, and testing.

**Beyond the Cranial Vault (BTCV) Dataset (Landman et al., 2015).** The BTCV dataset, released as part of the MICCAI 2015 Multi-Atlas Labeling Beyond the Cranial Vault challenge, is a widely used benchmark for abdominal organ segmentation. It consists of 50 contrast-enhanced abdominal CT scans provided by Vanderbilt University Medical Center, acquired in the portal venous phase from patients with either metastatic liver cancer or postoperative abdominal wall hernia. Each scan is annotated with 13 abdominal organs, including the liver, spleen, pancreas, kidneys, stomach, gallbladder, esophagus, aorta, inferior vena cava, and duodenum, among others. The scans exhibit variable field of view and resolution, with in-plane spacing ranging from $0.54 \times 0.54$ mm$^2$ to $0.98 \times 0.98$ mm$^2$ and slice thickness between 2.5 mm and 5.0 mm. In this study, we adopt 30 scans from the official training set as an external test set to evaluate the generalization performance of our model.

**Automatic Cardiac Diagnosis Challenge (ACDC) Dataset (Bernard et al., 2018).** The ACDC dataset comprises cardiac MRI scans collected over six years at the University Hospital of Dijon, acquired with 1.5T and 3.0T Siemens scanners. Each case includes short-axis cine sequences with expert annotations of the same anatomical structures as in M&Ms, at both end-systolic and end-diastolic phases. In our study, ACDC serves as an external benchmark to assess the generalization capability of our model, where we adopt the 200 official training cases as validation data and the 100 official test cases for final evaluation.

**University of Waterloo Skin Cancer (UW-SC) Dataset (University of Waterloo).** The UWaterloo Skin Cancer Dataset comprises 167 dermoscopic images of skin lesions, with manual segmentation masks provided for each lesion and verified by experts. In our work, we leverage this dataset to evaluate the segmentation performance in a challenging and heterogeneous clinical image collection.

**Breast Ultrasound Dataset B (BUS) (Al-Dhabyani et al., 2020).** BUS is a breast ultrasound collection designed for region-of-interest (ROI) detection and lesion localization. It contains 163 ultrasound images with expert-annotated lesion regions. In our study, all benign and malignant lesions are merged into a single lesion class to simplify the task. We use this dataset solely as an external validation set to assess the generalization capability of our model.

**BraTS Dataset (Baid et al., 2021).** The BraTS series of datasets is a widely used benchmark in brain tumor analysis. It provides multimodal MRI scans with expert annotations of glioma substructures. In our experiments, we test our method on the same 369 slices as in (Liu et al., 2023b).

**M&Ms-2 Dataset (Campello et al., 2021).** The Multi-Centre, Multi-Vendor & Multi-Disease (M&Ms-2) dataset was released as part of the M&Ms-2 challenge at MICCAI 2021. For the M&Ms-2 dataset, we utilize only the long-axis (LA) cine MRI images, which are not present in the M&Ms-1 dataset. This setting enhances data diversity and allows us to assess the generalization ability of our model under an unseen-class scenario. In our experiments, we exclusively use the 320 long-axis (LAX) cine MRI images from the official M&Ms-2 test set as an external test set to evaluate model generalization. The 400 LAX images from the official training set are adopted as a validation set, while none of the M&Ms-2 images are used during model training.

## A.1.2 DATASET STATISTICS

As shown in Table 5, we train and evaluate our framework on a diverse set of publicly available medical image segmentation datasets spanning multiple imaging modalities and anatomical regions. The training pool covers 12 datasets: abdominal CT/MRI (e.g., AMOS, LiTS, KiTS), cardiac MRI (M&Ms, ACDC, LAScarQS), ultrasound (Breast cancer), dermoscopy (ISIC), endoscopy (BKAI_POLY), pathology (KPIs), fundus imaging (PAPILA), and chest X-rays. These datasets vary substantially in size, with sample counts ranging from a few hundred to several thousand 3D volumes or 2-D slices, and in annotation granularity, from single-organ labels to multi-organ delineations. For evaluation, we adopt both external and unseen-class test sets to assess generalization across modalities, anatomical structures, and imaging centers.

Table 6: Datasets statistics.

| Dataset | # cls | Modality | 3D Volumes | | | 2-D Slices | | |
|---|---|---|---|---|---|---|---|---|
| | | | Train | Validate | Test | Train | Validate | Test |
| AMOS_CT | 13 | CT | 190 | 10 | 100 | 33944 | 1689 | 17861 |
| AMOS_MRI | 13 | MRI | 38 | 2 | 20 | 5160 | 263 | 2614 |
| M&Ms | 3 | MRI | 300 | - | 340 | 2475 | - | 2821 |
| LiTS | 1 | CT | 104 | - | 27 | 5687 | - | 1625 |
| KiTS | 1 | CT | 168 | - | 42 | 5269 | - | 1609 |
| LAScarQS | 1 | MRI | 98 | - | 32 | 3561 | - | 1183 |
| Chest X-ray | 1 | X_ray | - | - | - | 450 | - | 116 |
| Breast Cancer | 1 | Ultrasound | - | - | - | 518 | - | 129 |
| KPIs | 1 | Pathology | - | - | - | 8261 | 500 | 4055 |
| PAPILA | 1 | Fundus | - | - | - | 390 | - | 98 |
| BKAI_POLY | 2 | Endoscopy | - | - | - | 800 | - | 200 |
| ISIC | 1 | Dermoscopy | - | - | - | 2594 | 100 | 1000 |
| BTCV | 13 | CT | - | - | 30 | - | - | 3791 |
| ACDC | 3 | MRI | - | 200 | 100 | - | 1841 | 1001 |
| UW-SC | 1 | Dermoscopy | - | - | - | - | - | 167 |
| BUS | 1 | Ultrasound | - | - | - | - | - | 163 |
| BraTS | 1 | MRI | – | – | – | - | - | 369 |
| M&Ms-2 | 3 | MRI | – | 400 | 320 | - | 400 | 320 |

## A.2 IMPLEMENTATION

**Training details.** Our models are trained for 75 epochs with a batch size of 16 on 8 Quadro RTX 8000 GPUs using the AdamW optimizer with a base learning rate of $1 \times 10^{-4}$. A cosine annealing scheduler with 10 warm-up epochs is applied, where the minimum learning rate is scaled by $1 \times 10^{-5}$. Images are resized to $512 \times 512$ during training, with augmentations including random flips, affine transforms (shift, scale, rotation), brightness/contrast adjustments, Gaussian blur/noise, and grid distortions. To encompass all operational modes, each training batch is randomly assigned to one mode, with probabilities of 0.3, 0.3, and 0.4 for Mode-1, Mode-2, and Mode-3, respectively. During inference, input images are resized with the long side fixed to 512 while preserving aspect ratio, and Dice scores are computed after remapping predictions to the original resolution. We use a combination of binary cross-entropy loss and Dice loss to compute the mask loss $L = L_{ce} + L_{dice}$. The same settings are applied on our ablation study.

**Click representation.** In both training and inference, user clicks are encoded as disk-shaped maps with a fixed radius of 1 pixel. Consistent with previous studies (Sofiiuk et al., 2022; Liu et al., 2023b), simulated clicks are generated by comparing the predicted segmentation against the ground truth. Differing from prior strategies, however, we place each new click at the centroid of the largest misclassified connected component, which more closely mimics practical user interactions in medical image analysis. We consider three operating modes: in Mode-1 and Mode-2, the model produces an initial mask and then refines it using two clicks, whereas in Mode-3 the model iteratively applies three clicks, yielding three successive segmentation masks.

**Additional implementation details for baseline models.** For SAM2, which is pretrained on large-scale natural images, we initialize the model using the official `sam2.1_hiera_base_plus` checkpoint and fine-tune it on our curated medical datasets under the same training settings as all other methods. For SegNext and MedSAM, the ViT-based encoder is first initialized using MAE-pretrained ViT-Base weights. After loading these pretrained encoder weights, the entire network is jointly trained end-to-end on our curated medical datasets. For other medical segmentation models such as MultiverSeg, to avoid any potential data overlap with their original training data sources, we train them from scratch on our curated datasets to ensure fair comparison and full convergence. User interactions (positive/negative clicks) are simulated consistently across all methods following the same policy described above.

### A.3 MODEL ARCHITECTURE

**Image encoder.** We employ UNet (Isensee et al., 2021) as the image encoder, a widely adopted lightweight backbone for medical image segmentation. The encoder extracts hierarchical features at three different resolutions, namely $1/16$, $1/8$, and $1/4$ of the original image, yielding a comprehensive multi-scale representation ($S = 3$). In our experiments, the feature maps have spatial sizes of $32 \times 32$, $64 \times 64$, and $128 \times 128$, with channel dimensions of 384, 192, and 96, respectively.

**Mask encoder & Interactive prompt encoder.** We utilize a lightweight mask encoder to extract hierarchical representations from the reference masks. The mask encoder employs simple consecutive convolutional blocks to generate hierarchical features that are spatially aligned with those from the image encoder, producing reference mask features at the same scales ($1/4$, $1/8$, and $1/16$ of the input resolution). For the interactive prompt encoder, we adopt an identical architectural design to ensure consistency.

**Fusion of image features and 2-D dense prompts.** After extracting features from the image encoder and either the mask encoder or the interactive prompt encoder, we employ a unified prompt fusion module to combine them. To fully exploit contextual cues, our model operates on multi-scale features. For clarity, the main paper illustrates the architecture with a single-scale example, while the appendix provides the details of the multi-scale processing. Specifically, the feature maps $\{\boldsymbol{F}_{32}, \boldsymbol{F}_{64}, \boldsymbol{F}_{128}\}$ are obtained from the image encoder at different resolutions.

For Mode-3 (interactive segmentation), fusion is straightforward: the image features are directly added to the interactive click and mask features, as they share the same dimensionality. For Mode-2, as discussed earlier, fusion is applied only at the lowest resolution $\boldsymbol{F}_{32}$ ($384 \times 32 \times 32$). The resulting fused features are then propagated across higher scales using the residual connections described in the following section.

After flattening, we define the reference keys, values, and query keys as:
$$\boldsymbol{K}^{\text{ref}} = \text{Flat}(\boldsymbol{F}_k) \in \mathbb{R}^{C \times N_{\text{ref}}hw}, \quad \boldsymbol{V}^{\text{ref}} = \text{Flat}(\boldsymbol{F}_v) \in \mathbb{R}^{C \times N_{\text{ref}}hw}, \quad \boldsymbol{K}^q = \text{Flat}(\boldsymbol{F}_q) \in \mathbb{R}^{C \times hw},$$
(12)

where $h = w = 32, C = 384$.

For any similarity function $c : \mathbb{R}^C \times \mathbb{R}^C \to \mathbb{R}$, the pairwise affinity matrix $\boldsymbol{A}$ and its softmax-normalized form $\boldsymbol{W}$ are computed as:
$$\boldsymbol{A}_{i,j} = c\big(\boldsymbol{K}^{\text{ref}}_{:,i}, \boldsymbol{K}^q_{:,j}\big), \quad \boldsymbol{W}_{i,j} = \frac{\exp(\boldsymbol{A}_{i,j})}{\sum_n \exp(\boldsymbol{A}_{n,j})}, \quad \boldsymbol{A}, \boldsymbol{W} \in \mathbb{R}^{N_{\text{ref}}hw \times hw}.$$
(13)

Following Cheng et al. (2021; 2024), we adopt the negative squared Euclidean distance to compute $\boldsymbol{A}$:
$$\boldsymbol{A}^{\text{L2}}_{i,j} = -\big\|\boldsymbol{K}^{\text{ref}}_{:,i} - \boldsymbol{K}^q_{:,j}\big\|_2^2 = 2\boldsymbol{K}^{\text{ref}}_{:,i}\boldsymbol{K}^q_{:,j} - \big\|\boldsymbol{K}^{\text{ref}}_{:,i}\big\|_2^2 - \big\|\boldsymbol{K}^q_{:,j}\big\|_2^2,$$
(14)

where the last term can be omitted as shown in (Cheng et al., 2021), which improves efficiency and reduces computational cost. Finally, the 2-D fusion feature map is aggregated as:

$$\boldsymbol{F}_{\text{fuse}} = \boldsymbol{V}^{\text{ref}}\boldsymbol{W}, \quad \boldsymbol{F}_{\text{fuse}} \in \mathbb{R}^{C \times hw}, \tag{15}$$

and is passed into the decoder for mask prediction. To facilitate subsequent operations, the feature channels at all scales are projected to 256 via a linear layer before being fed into the decoder.

**Generating the click queries.** In this section, we provide additional details on how user clicks are encoded into 1-D query embeddings for interactive segmentation. Given a user click at pixel coordinates $P_0 = (x, y)$ in the original image $\boldsymbol{I} \in \mathbb{R}^{3 \times H \times W}$, we first map it to the downsampled feature space $\boldsymbol{F}^s \in \mathbb{R}^{C \times (H/s) \times (W/s)}$, where $s$ denotes the stride of the encoder. The corresponding coordinates are:

$$x' = \left\lfloor \tfrac{x}{s} \right\rfloor, \quad y' = \left\lfloor \tfrac{y}{s} \right\rfloor. \tag{16}$$

Around the mapped point $(x', y')$, we extract a $(2r + 1) \times (2r + 1)$ local window to capture neighborhood context. We set $r = 1$ in our experiments. The pooled feature vector is then obtained via average pooling:

$$\boldsymbol{f}_{\text{pooled}} = \frac{1}{(2r + 1)^2} \sum_{i=-r}^{r} \sum_{j=-r}^{r} \boldsymbol{F}^s_{x'+i, y'+j}. \tag{17}$$

The pooled feature is projected into the query space using a multilayer perceptron (MLP):

$$\boldsymbol{x}_f = \text{MLP}(\boldsymbol{f}_{\text{pooled}}), \quad \boldsymbol{x}_f \in \mathbb{R}^{1 \times C}. \tag{18}$$

To obtain the final single click query embedding, we incorporate the positional encoding from SAM (Kirillov et al., 2023):

$$\boldsymbol{q}^c = \boldsymbol{x}_f + PosEmbed(x, y), \quad \boldsymbol{q}^c \in \mathbb{R}^{1 \times C}. \tag{19}$$

We generate two separate groups of click queries, $\boldsymbol{Q}^{\text{pos}}$ for positive clicks and $\boldsymbol{Q}^{\text{neg}}$ for negative clicks which correspond to the positive and negative attention masks. Since the number of clicks in the two groups may differ during interaction, the smaller group is padded with dummy queries (zero vectors) to maintain balance. This design ensures stable training and inference when combining click queries across interaction steps. The two groups are then concatenated into the final 1-D click queries $\boldsymbol{Q}^c$.

**Decoder**. As shown in Figure 2 (b), we use $L = 6$ layers in total in the transformer decoder. Similar to Mask2Former (Cheng et al., 2022), we adopt a round-robin strategy for multi-scale interaction between image features and integrated queries. The blocks are scheduled in a cyclic order across scales (i.e., $1 \rightarrow 2 \rightarrow 3 \rightarrow 1 \rightarrow 2 \rightarrow 3$), ensuring balanced cross-scale information exchange throughout the decoding process. Finally, the decoded features are passed through a lightweight mask decoder that projects them back to the spatial resolution of the input, producing the final segmentation masks.

**Residual connections across scales.** Following Verse (Guo et al., 2025), we adopt residual resampling connections to enhance the interaction between 2-D fusion features and 1-D queries across multiple scales. Concretely, for a feature map $\boldsymbol{F}_l$ at layer $l$, we first resample it to match the resolution of the next layer $\boldsymbol{F}_{l+1}$:

$$\boldsymbol{F}_l^{\text{resampled}} = \text{Resample}\left(\boldsymbol{F}_l\right), \tag{20}$$

where Resample$(\cdot)$ denotes upsampling or downsampling depending on the relative scales. We then compute a residual representation via a convolutional layer and add it to the next-layer features:

$$\boldsymbol{F}_{l+1}^{\text{updated}} = \boldsymbol{F}_{l+1} + \text{Res}\left(\boldsymbol{F}_l^{\text{resampled}}\right). \tag{21}$$

By iteratively resampling and aggregating residuals across layers, the model improves the interaction between features and queries over different scales.

**Foreground-background masked attention.** Attention masks have been shown to improve the efficiency of attention by constraining it to more relevant regions (Cheng et al., 2022; 2024). Therefore, we employ attention masks in every decoder block to consistently guide query updating and

feature refinement. We take in-context segmentation as an example to illustrate how attention masks are used. In this setting, we obtain a 1-D query set $\boldsymbol{Q}^s \in \mathbb{R}^{n_s \times C}$, where the first $\frac{n_s}{2}$ queries correspond to the foreground and the remaining $\frac{n_s}{2}$ queries correspond to the background. The queries are then updated through a foreground-background masked attention mechanism. This design enables multiple queries to focus on target-related image features while also leveraging complementary background information.

The foreground-background masked cross-attention at the $l$-th decoder layer is formulated as:

$$\boldsymbol{Q}'_l = \text{softmax}(\boldsymbol{M}_l + \boldsymbol{Q}_l \boldsymbol{K}_l^\top)\boldsymbol{V}_l + \boldsymbol{Q}_l, \tag{22}$$

where $\boldsymbol{Q}_l$ denotes the transformed queries at layer $l$, and $\boldsymbol{K}_l, \boldsymbol{V}_l$ are the linear projections of the key features $\boldsymbol{F}_l$. We omit the standard $\frac{1}{\sqrt{C}}$ scaling term for brevity. The binary mask $\boldsymbol{M}_l \in \{0, -\infty\}^{n_s \times h_l w_l}$ is used to enforce the foreground-background separation. At each layer $l$, we first predict a probability mask $\boldsymbol{P}_l \in [0, 1]^{h_l \times w_l}$, where each value represents the foreground probability at spatial location $(i, j)$, which is obtained from the pixel features $\boldsymbol{F}_{l-1}$ of the previous layer using a lightweight mask decoder followed by resizing. The n-th mask $\boldsymbol{M}_l(i, j)$ at spatial location $(i, j)$ is defined as:

$$\boldsymbol{M}_l(i,j) = \begin{cases} 0, & \boldsymbol{P}_l(i,j) \geq 0.5 \text{ and } n < \frac{n_s}{2}, \\ 0, & \boldsymbol{P}_l(i,j) < 0.5 \text{ and } n \geq \frac{n_s}{2}, \\ -\infty, & \text{otherwise.} \end{cases} \tag{23}$$

The same procedure is applied in mode-3 interactive segmentation, where positive and negative click queries represent the foreground and background masks, respectively. When the number of positive and negative clicks is unbalanced, dummy clicks are added to equalize the query set. For semantic segmentation, only the foreground (positive) mask is applied.

**Other hyperparameters.** For semantic segmentation, the number of 1-D queries is set to $p = 2$, while in context segmentation we use $n_s = 6$. We observe only minor performance differences when varying the query number. In the transformer decoder, all feature channels are fixed at 256, and each attention block uses 8 heads. We set the number of experts $M = 5$ in our main experiments. During training, single image–mask pair is employed as the reference.

## A.4  MORE EXPERIMENTS

**Analysis of expert weights.** Figure 4 illustrates the distribution of softmax expert weights across different modes when tested on external ACDC dataset. We observe that the weighting patterns vary substantially by mode, reflecting the task-specific routing behavior of the MoE decoder. In Mode-1, Expert 5 receives the highest weight, suggesting specialization toward that expert, while the other experts contribute more evenly. In contrast, Mode-2 strongly favors Expert 1, with minimal reliance on Experts 4 and 5, indicating a different specialization strategy. Mode-3 shows a more balanced distribution across all experts, with moderate weights assigned consistently. These results confirm that the model dynamically allocates expert capacity depending on the input mode, which supports the effectiveness of the MoE design in capturing heterogeneous segmentation requirements.

**Analysis of convergence curves of all datasets.** Figure 5 shows how interactive segmentation (Mode-3) performance of K-Prism changes as the number of clicks increases across all datasets. In most cases, accuracy improves quickly with just a few clicks and then gradually levels off, which highlights the benefit of interactive refinement. Our model consistently achieves the strongest results on the majority of datasets, keeping a clear advantage over baselines across the entire interaction range. We also note some variation: certain datasets reach high accuracy after only a few clicks, while others need more interactions to stabilize, likely due to differences in image quality, structural complexity, or segmentation difficulty.

**Analysis of multi-mode training.** Table 7(a) compares joint training with single-mode training under the same ablation setting as used in the main paper, where models are trained on the M&Ms and Breast Cancer dataset and evaluated on ACDC and BUS. We find that our unified framework achieves performance on par with or better than training each mode individually, while being much more efficient since a single model handles all paradigms simultaneously. This indicates that knowledge sharing across modes is beneficial: semantic and in-context learning provide complementary

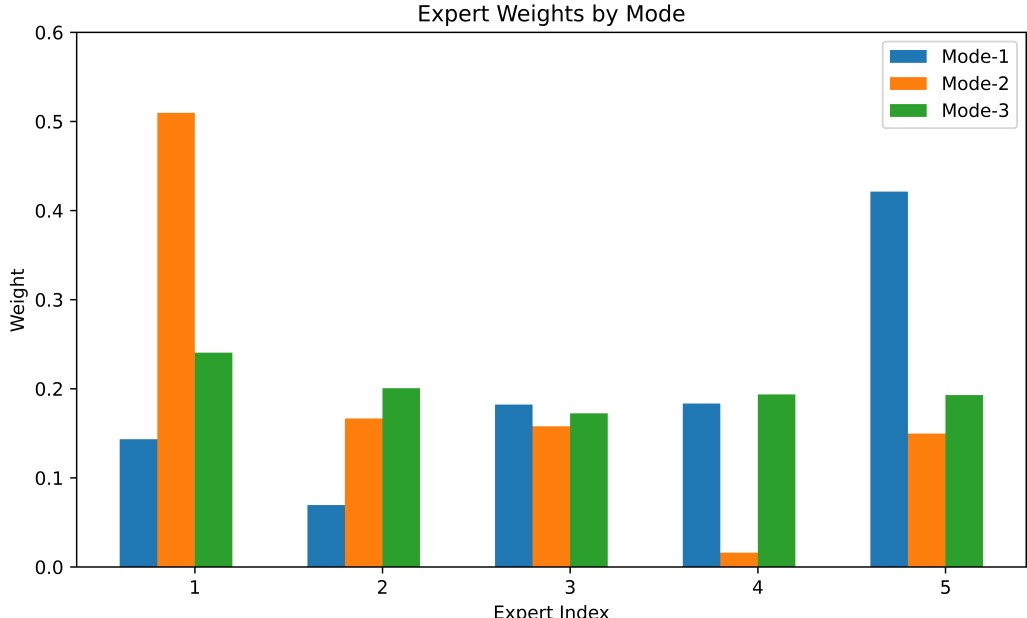

Figure 4: Distribution of softmax expert weights across different modes on the external ACDC dataset.

Table 7: Extensive ablation studies.

(a) Joint training achieves performance comparable to single-mode training, while being three times more efficient.

| Method | Semantic | In-context | Interactive | | | |
|---|---|---|---|---|---|---|
| | Dice | Dice | NoC90 ↓ | NoC95 ↓ | Dice(1) ↑ | Dice(5) ↑ |
| Ours | 81.28 | 79.21 | 2.31 | 4.80 | 86.76 | 93.79 |
| Mode-1 only | 80.57 | - | - | - | - | - |
| Mode-2 only | - | 80.84 | - | - | - | - |
| Mode-3 only | - | - | 2.34 | 4.97 | 86.55 | 93.68 |

(b) As the number of experts increases, model performance improves, albeit at the cost of a larger number of parameters.

| # Expert | # Params | Semantic | In-context | Interactive | | | |
|---|---|---|---|---|---|---|---|
| | | Dice | Dice | NoC90 ↓ | NoC95 ↓ | Dice(1) ↑ | Dice(5) ↑ |
| 5 (Ours) | 43.29M | 81.28 | 79.21 | 2.31 | 4.80 | 86.76 | 93.79 |
| 4 | 39.34M | 79.80 | 77.97 | 2.38 | 4.96 | 86.02 | 93.43 |
| 3 | 35.39M | 79.24 | 76.81 | 2.58 | 5.03 | 84.24 | 92.70 |
| 2 | 31.43M | 77.62 | 75.55 | 2.64 | 5.12 | 84.16 | 92.22 |

supervision that improves overall representation quality, and interactive refinement benefits from the shared backbone. These results highlight the practicality of our unified design, which avoids training separate models yet still delivers competitive or superior accuracy across different segmentation settings.

**Analysis of number of experts.** Table 7(b) analyzes the effect of varying the number of experts in the MoE decoder. We observe a consistent performance improvement as the number of experts increases from 2 to 5 across semantic, in-context, and interactive settings. Dice scores steadily rise while the number of clicks required to reach target accuracy decreases. These gains, however, come with larger parameter count and additional computational overhead. Due to resource constraints, we were unable to scale beyond five experts, but the trend suggests that larger expert capacity may yield further benefits. This indicates strong potential for scaling K-Prism into even more powerful seg-

Table 8: Throughput (FPS) with different numbers of experts in MoE decoder on a single A100 GPU.

| # Experts | 5 | 4 | 3 | 2 | 1 (Plain) |
|-----------|------|------|------|------|-----------|
| FPS | 3.63 | 4.24 | 4.61 | 5.10 | 5.38 |

Table 9: Sensitivity of in-context segmentation to reference exemplar selection. Results are averaged over 10 independent 1-shot evaluations per dataset.

| Dataset | Mean | Std | Min | Max |
|---------|-------|------|-------|-------|
| ACDC | 83.68 | 2.38 | 79.10 | 86.18 |
| BUS | 78.43 | 0.72 | 77.09 | 79.15 |

mentation models, potentially improving initialization robustness and reducing annotation workload in clinical workflows. To further quantify inference efficiency, we evaluate throughput on the ACDC dataset using a single A100 GPU across different numbers of experts (Table 8). The results show that increasing the number of experts introduces only moderate overhead: the throughput decreases by roughly 1.7 FPS (frames per second) when moving from a plain decoder (1 expert) to a 5-expert MoE decoder. Importantly, in practical clinical deployments that typically operate on multi-GPU servers, throughput scales nearly linearly with available GPUs, meaning that adding more experts does not compromise real-time usability.

**Analysis of reference exemplar sensitivity in in-context segmentation.** To further assess the robustness of the in-context segmentation mode (Mode-2), we evaluate how performance varies with different choices of reference exemplars. We conduct experiments on two representative external datasets: BUS (2D ultrasound) and ACDC (3D cardiac MRI). For each dataset, we randomly select 10 distinct reference exemplars (independent subjects/images). Using each exemplar as the sole reference, we run one-shot in-context segmentation on all query images, yielding 10 independent evaluations per dataset. Table 9 reports the resulting mean, standard deviation, and minimum–maximum range across these evaluations. The BUS dataset shows extremely low variance ($78.43 \pm 0.72$), indicating that Mode-2 predictions remain highly stable across different reference choices in typical 2D settings. For ACDC, where anatomical and positional variability across subjects is inherently larger, the variance is somewhat higher ($83.68 \pm 2.38$) but still falls within a moderate and acceptable range. Overall, these results demonstrate that K-Prism is robust to reasonable variations in reference exemplar quality and does not depend on a highly specific exemplar to achieve strong in-context segmentation performance.

**Analysis of reference quantity for few-shot in-context segmentation.** To further examine the Mode-2 bottleneck in complex multi-organ scenarios, we conduct a few-shot in-context study on the AMOS_MRI dataset. Because axial slices in 3D multi-organ volumes exhibit substantial anatomical variability, a single reference exemplar may misalign with many target slices. To assess whether additional exemplars can mitigate this issue, we evaluate 1-, 3-, 5-, and 9-shot settings, where reference slices are sampled from fixed axial locations corresponding to typical anatomical positions of the target organ within the 3D volume (1-shot: 50%; 3-shot: 20/50/80%; 5-shot: 20/40/50/60/80%; 9-shot: 10–90% at 10% intervals). As shown in Table 10, accuracy improves consistently with more exemplars, rising from 78.21% (1-shot) to 79.90% (9-shot). This shows that the performance drop observed under strict 1-shot conditions is not an inherent limitation of our framework: when additional exemplars are provided, K-Prism can effectively exploit them. These results highlight the adaptability and robustness of K-Prism's in-context mode in challenging multi-organ segmentation scenarios.

## A.5 VISUALIZATION

Figures 6,7 and 8 present qualitative comparisons of semantic, in-context, and interactive segmentation across diverse datasets and modalities. For semantic segmentation (Figure 6), K-Prism produces more accurate and consistent results than competing methods such as Clip-driven, UniSeg, and Hermes, particularly on challenging tumor and pathology cases. For in-context segmentation (Figure 7), K-Prism achieves clearer boundaries and higher Dice scores across both CT/MRI (e.g.,

Table 10: Few-shot in-context segmentation results on AMOS_MRI. Increasing the number of reference exemplars improves performance in multi-organ scenarios.

| Shots | 1 | 3 | 5 | 9 |
|---|---|---|---|---|
| Dice (%) | 78.21 | 79.22 | 79.70 | 79.90 |

AMOS, M&Ms) and endoscopic datasets (e.g., BKAI_POLY). For interactive segmentation (Figure 8), we visualize results at the fifth click, showing that K-Prism converges faster and yields more precise masks compared with strong baselines such as nnInteractive, MultiverSeg, SAM2, and Seg-Next. Overall, these visualizations demonstrate the robustness and versatility of K-Prism across three segmentation paradigms.

### A.6 ANALYSIS OF FAILURE CASES FOR K-PRISM

In the BraTS example (Figures 9, top-left), the model mistakenly segments the bright peritumoral edematous/invaded tissue (Yousef et al., 2023) as tumor. Because the reference image contains a rounded, high-intensity tumor with a compact mass-like morphology and the peritumoral edematous/invaded tissue signal in the query slice exhibits a superficially similar intensity distribution, the appearance-driven matching in Mode-2 incorrectly aligns this region with the tumor exemplar. Crucially, brain MRI is a completely unseen domain for our model: no brain anatomy, texture patterns, or disease manifestations appear in the training datasets. As a result, the model lacks any semantic prior to distinguish true oncologic tissue from peritumoral edematous/invaded tissue hyperintensities. Under such severe domain and anatomical shifts, Mode-2 naturally over-relies on low-level intensity correspondences in the absence of domain knowledge, causing misleading but internally consistent matches.

In the M&Ms-2 example (Figures 9, bottom-left), the model incorrectly segments part of the left atrial wall as left ventricle myocardium. M&Ms-2 long-axis cardiac MRI has never appeared in the training data, and its anatomical geometry differs substantially from the short-axis cardiac MRI datasets used for model development. Importantly, the left atrial wall and the left ventricular myocardium exhibit highly similar shaped appearances in the long-axis view. Because Mode-2 relies heavily on appearance-based alignment between the reference exemplar and the query slice, the model is misled by this shape similarity. As a result, the model aligns the atrial wall with the "myocardium" region from the exemplar, despite the anatomical mismatch.

### A.7 THE USE OF LARGE LANGUAGE MODELS (LLMS)

In this work, large language models (LLMs) were used solely as general-purpose writing assistants for text polishing and formatting adjustments. They did not contribute to research ideation, experimental design, analysis, or result interpretation.

### A.8 ETHICS STATEMENT

This research adheres to the ICLR Code of Ethics and its guiding principles of responsible stewardship, fairness, and transparency. All experiments are conducted exclusively on publicly available, de-identified datasets. No identifiable patient data were collected or generated, and all datasets include appropriate ethical approvals from their original organizers. Our goal is to advance trustworthy and socially beneficial AI for medical image analysis. While the proposed framework demonstrates strong performance, it is intended for research use only and not for direct clinical deployment without further validation. We acknowledge that premature use of automated segmentation could introduce risks, including potential bias across populations, imaging protocols, or disease types. To mitigate these concerns, we evaluate extensively across diverse datasets and modalities and highlight limitations in out-of-distribution generalization. We also stress the importance of fairness, reproducibility, and responsible application in any future clinical translation.

## A.9 REPRODUCIBILITY STATEMENT

We take reproducibility seriously and provide extensive details throughout the main text and Appendix, including dataset descriptions, preprocessing steps, model architecture, training hyperparameters, and evaluation protocols. Comprehensive ablation studies are included to highlight the contribution of each component. All experiments are conducted on publicly available datasets. The full codebase is publicly available to support replication and further research.

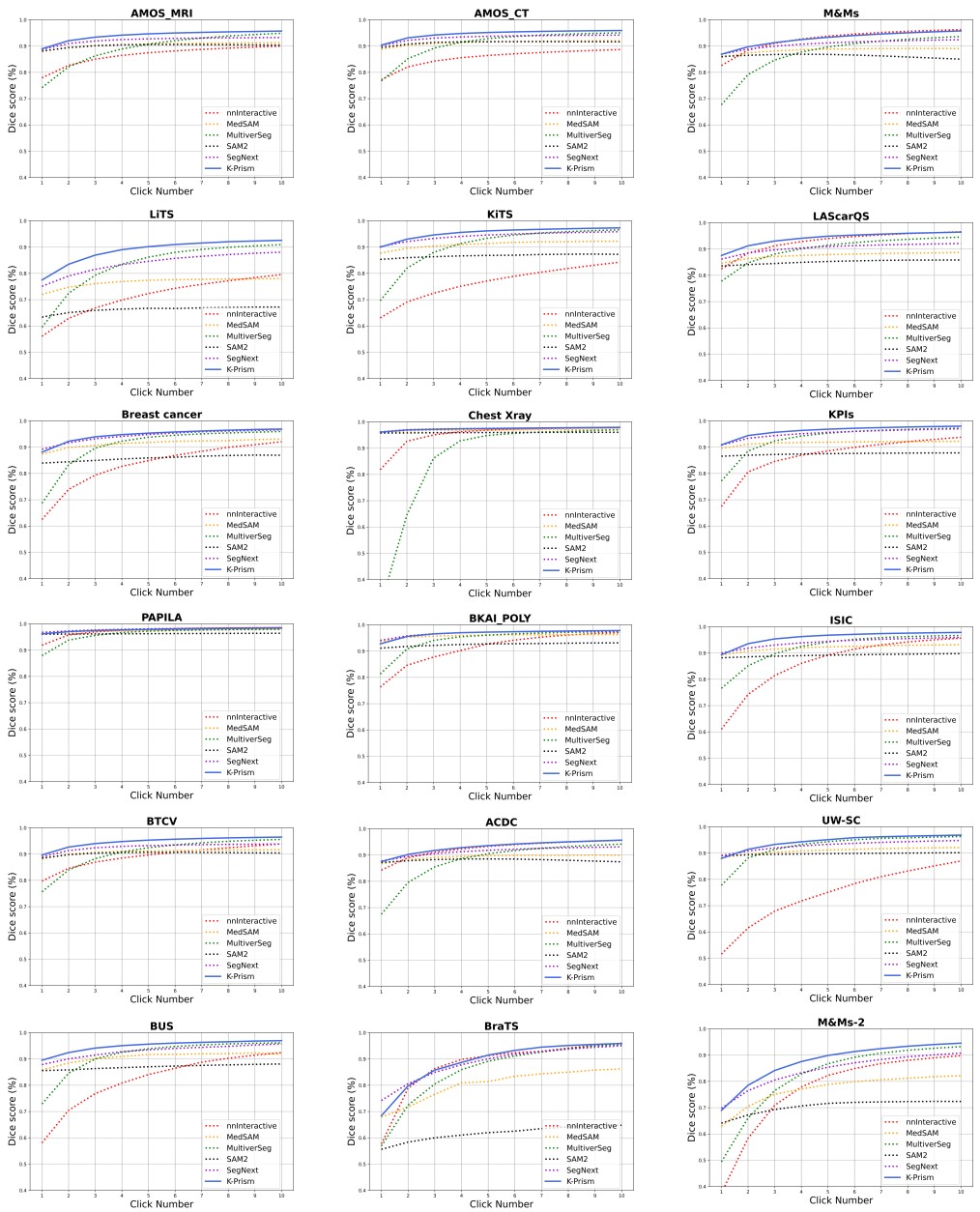

Figure 5: Convergence curves of K-Prism's interactive segmentation (Mode-3) on all datasets.

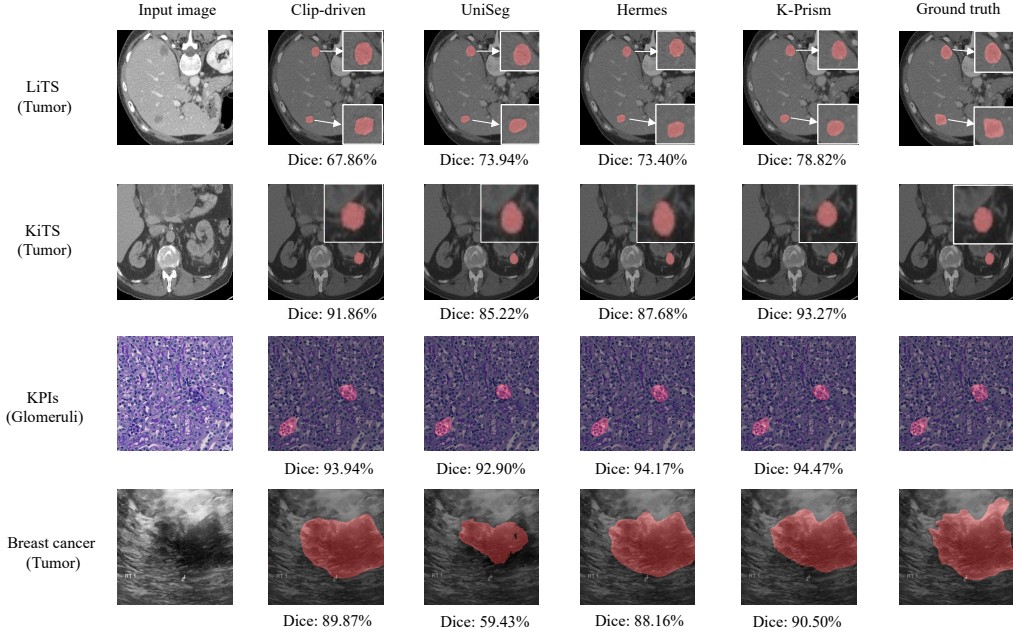

Figure 6: Qualitative comparison of semantic segmentation across representative datasets: LiTS (tumor), KiTS (tumor), KPIs (glomeruli), and Breast cancer (tumor). From left to right: input image, predictions from universal models Clip-driven, UniSeg, Hermes, and K-Prism, and the ground truth. Reported Dice scores highlight that K-Prism produces more accurate and consistent results across diverse modalities and targets.

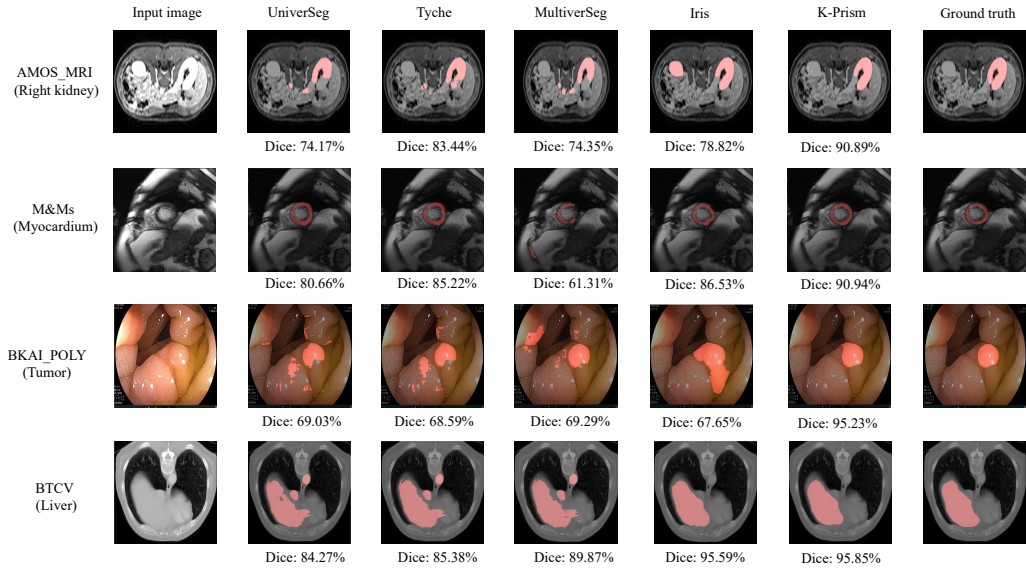

Figure 7: Qualitative comparison of in-context segmentation across four representative datasets: AMOS_MRI (right kidney), M&Ms (myocardium), BKAI_POLY (tumor), and BTCV (liver). From left to right: input image, predictions from UniverSeg, Tyche, MultiverSeg, Iris, and K-Prism, and the ground truth. Reported Dice scores show that K-Prism achieves the most accurate and consistent results across diverse modalities and targets.

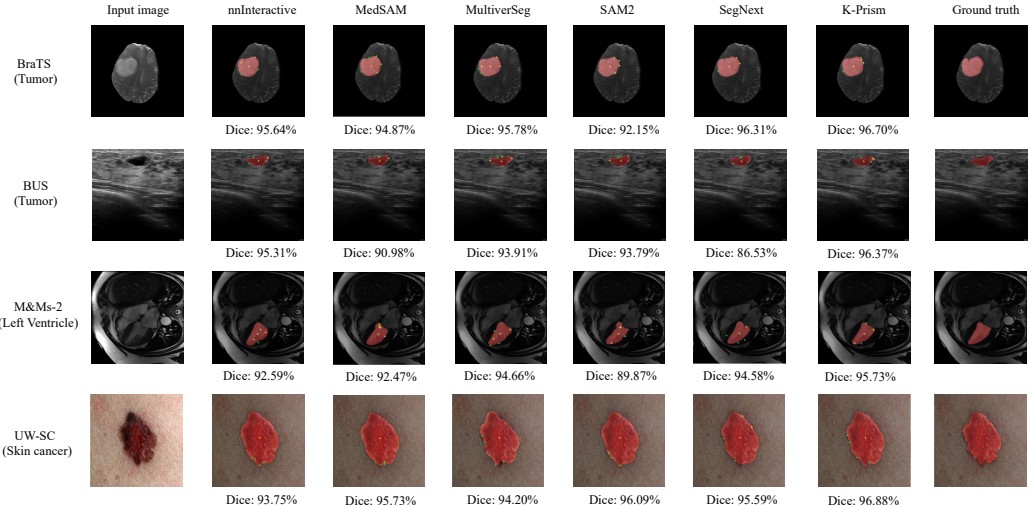

Figure 8: Qualitative comparison of interactive segmentation results at the fifth click across four representative datasets: BraTS (tumor), BUS (tumor), M&Ms-2 (left ventricle), and UW-SC (skin cancer). From left to right: input image, predictions from nnInteractive, MedSAM, MultiverSeg, SAM2, SegNext, and K-Prism, and the ground truth. Reported Dice scores show that K-Prism consistently produces the most accurate and reliable segmentations across diverse modalities and targets.

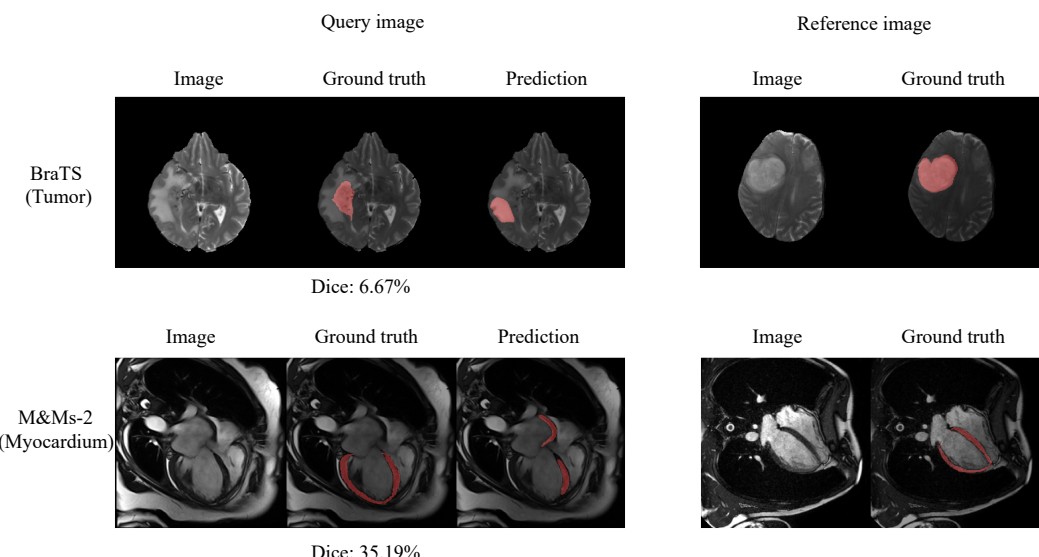

Figure 9: Representative failure cases of in-context segmentation (Mode-2) on unseen datasets: BraTS and M&Ms-2.

