# OpenReview forum: "K-Prism: A Knowledge-Guided and Prompt Integrated Universal Medical Image Segmentation Model"
_ICLR.cc/2026/Conference — ICLR 2026 Poster_

### Official Review · Reviewer_GsDa · 2025-10-31

**Soundness:** 3
**Presentation:** 3
**Contribution:** 3
**Rating:** 8
**Confidence:** 4

**Summary:**

This work propose K-Prism, which is a unified framework for medical image segmentation that innovatively integrates three knowledge paradigms into a single model: semantic priors (Mode-1), in-context exemplars (Mode-2), and interactive feedback (Mode-3). Its core technique encodes heterogeneous knowledge into a unified dual-prompt representation (1D sparse prompts for "what" and 2D dense prompts for "where") , which is then processed by a Mixture-of-Experts (MoE) decoder with dynamic routing. Experiments on 18 public, multi-modal datasets demonstrate SOTA performance across all three segmentation settings.

**Strengths:**

- Significant Problem: Directly addresses the long-standing problem of model fragmentation in medical segmentation.

- Novel Architecture: The proposed "dual-prompt" and MoE decoder design is elegant, successfully unifying three heterogeneous knowledge sources.

- Comprehensive Experiments: Extensively validated on 18 public datasets covering diverse modalities.

- SOTA Performance: Outperforms existing state-of-the-art methods in all three tracks: semantic, in-context, and interactive segmentation.

**Weaknesses:**

- 2D Limitation: The model currently processes 3D medical data (like CT/MRI) based on 2D slices, which loses 3D contextual information.
- In-context Bottleneck: For complex multi-organ segmentation tasks, the in-context mode (Mode-2) performs noticeably worse than the semantic mode (Mode-1).
- Computational Cost: The MoE architecture, while effective, introduces a higher parameter count and computational overhead, potentially hindering real-time deployment.

**Questions:**

- 3D Extension: What are the main challenges in extending this framework to 3D? Specifically, can the dual-prompt and MoE mechanisms be efficiently scaled computationally?

- In-context Bottleneck: Could using k-shot (k>1) references in Mode-2 help alleviate the performance drop observed in multi-organ tasks?

- MoE Expert Specialization: What specific functions have the MoE experts specialized in? (e.g., are some experts dedicated to positive clicks and others to negative clicks?)

- Reference Quality: How sensitive is the performance of Mode-2 (in-context) to the quality of the reference exemplar (e.g., inaccurate mask) or domain shift (e.g., different scanner)?

---

> ### Author Response · Authors · 2025-11-20
> **Response to Reviewer GsDa**
>
> We thank the reviewer for the constructive and encouraging assessment of our work. We appreciate the recognition of K-Prism’s unified design, dual-prompt architecture, and strong empirical performance across diverse datasets. The reviewer’s insightful comments and questions are highly valuable, and we address each concern in detail below.
>
> ## 1.Response to Weakness1 and Question1
>
> We thank the reviewer for this clinically meaningful question which also mentioned by Reviewer ZEBx.
>
> First, our framework can be naturally extended to 3D volumetric segmentation, as both the dual-prompt formulation and MoE decoder are dimension-agnostic. Converting from 2D to 3D merely involves substituting 2D patch embedding and convolutions with their 3D counterparts, without any structural or training modifications. While we acknowledge that 3D processing naturally introduces higher computational cost, this can be effectively mitigated by removing multi-scale feature branches (e.g., only using features with lowest resolution) or using a lighter MoE decoder, allowing K-Prism to scale to 3D volumes without substantial redesign.
>
> Second, our focus on 2D segmentation reflects real-world clinical workflows, as highlighted by insights from our clinical collaborators indicating that radiologists mainly perform boundary refinement on 2D slices, where fine-grained lesion or tumor boundaries can be precisely adjusted. Fully 3D interaction is less practical, as volumetric rendering can obscure internal structures. While semantic and in-context segmentation can be readily extended to 3D, the interactive mode remains most clinically meaningful in 2D, motivating our slice-based motivations and experiments.
>
> We may also explore a full 3D extension in future work. In particular, we plan to investigate 2D-to-3D propagation, which uses our efficient 2D interactive refinement to update the entire 3D volume. A concise version of this discussion has been incorporated into the revised Discussion section.
>
> ## 2. Response to Weakness2 and Question2
>
> We appreciate the reviewer’s insightful question regarding the Mode-2 bottleneck on complex multi-organ datasets. To further examine this issue, we conducted an additional few-shot experiment on AMOS_MRI in which we increased the number of reference exemplars from 1 to 9. We evaluated 1-shot, 3-shot, 5-shot, and 9-shot settings. Results are shown below:
>
> ### Few-shot in-context segmentation results
>
> | Shots | 1     | 3     | 5     | 9     |
> |-------|-------|-------|-------|-------|
> | Dice (%) | 78.21 | 79.22 | 79.70 | 79.90 |
>
> As shown, the performance improves steadily as more reference slices are provided. 9-shot inference yields a 1.7% Dice improvement over the 1-shot setting. These results demonstrate that incorporating multi-slice exemplars effectively mitigates the drop observed in complex multi-organ and multi-level volumes.
>
> This improvement is expected because:
> • Axial levels in 3D vary substantially; a single exemplar may mismatch the target slice.
> • Structural similarity across different organs can confuse single-exemplar matching.
>
> By supplying exemplars from multiple positions, the model gains richer contextual cues and better disambiguates similar anatomy. This highlights the flexibility and robustness of K-Prism beyond what single-exemplar baselines can achieve.
>
> ---
>
> ## 3. Response to Weakness3
>
> We thank the reviewer for highlighting the importance of inference efficiency in clinical workflows. To quantify this, we measured per-image throughput on the ACDC dataset using a single A100 GPU with varying numbers of MoE experts.
>
> ### Throughput (FPS) with different MoE expert numbers
>
> | # Experts        | 5    | 4    | 3    | 2    | 1 (Plain) |
> |------------------|------|------|------|------|-----------|
> | FPS              | 3.63 | 4.24 | 4.61 | 5.10 | 5.38      |
>
> These results demonstrate that the MoE decoder moderately reduces inference speed. However, it delivers substantial and consistent performance improvements, as evidenced in Table 7(b). In real clinical deployments—usually on multi-GPU servers—throughput scales linearly with available GPUs, meaning that adding more experts does not compromise real-time usability. More details are included in Appendix A.4.

---

> ### Author Response · Authors · 2025-11-20
> **Response to Reviewer GsDa**
>
> ## 4. Response to Question3
>
> We thank the reviewer for this insightful question. We have already included an analysis of expert specialization in Appendix A.4 in the first manuscript, where Figure 4 in Appendix reports the softmax weight distribution of all experts across the three operation modes on the external ACDC dataset. As shown, different modes trigger distinct routing patterns. These observations confirm that the MoE decoder learns task-aware specialization rather than collapsing to a single shared expert. For more details, please refer to the ``Analysis of Expert Weights'' subsection in Appendix A.4.
>
> ## 5. Response to Question4
>
> To assess how sensitive Mode-2 (in-context) segmentation is to reference exemplar quality, we conducted an additional experiment across ACDC (MRI) and BUS (ultrasound). For each dataset, we randomly selected 10 reference exemplars and performed a full one-shot evaluation using each exemplar as the only reference.
>
> ### Sensitivity analysis of Mode-2 for reference exemplar choice
>
> | Dataset | Mean  | Std   | Min   | Max   |
> |---------|-------|-------|-------|-------|
> | BUS     | 78.43 | 0.72  | 77.09 | 79.15 |
> | ACDC    | 83.68 | 2.38  | 79.10 | 86.18 |
>
> BUS shows extremely low variance, indicating stable Mode-2 behavior in typical 2D settings. ACDC shows slightly larger variation due to more complex 3D anatomical differences across subjects, yet the range remains moderate. These results demonstrate that K-Prism does not rely on highly specific exemplars and is robust to reasonable reference variation.
>
> We hope this analysis addresses the reviewer’s concerns and supports the practical reliability of our in-context segmentation design.
>
> ## 6. Summary
> We sincerely thank the reviewer for the constructive and thoughtful feedback. Your comments helped us strengthen the clarity, experimental depth, and clinical relevance of the work. In response to all raised concerns including 3D extension, few-shot in-context performance, inference efficiency, expert specialization, and sensitivity to reference exemplars, we provided additional experiments, detailed analyses, and expanded discussions in both the main text and the Appendix. We hope our clarifications and new evidence satisfactorily address all questions and that the reviewer finds the revised version significantly improved.

---

### Official Review · Reviewer_3nT9 · 2025-10-31

**Soundness:** 1
**Presentation:** 2
**Contribution:** 1
**Rating:** 2
**Confidence:** 5

**Summary:**

The paper proposes K-Prism, a unified medical image segmentation framework that integrates three knowledge sources—semantic priors, few-shot in-context examples, and interactive user feedback. Using a dual-prompt and Mixture-of-Experts design, K-Prism flexibly adapts across tasks and modalities. Tested on 18 public datasets, it achieves state-of-the-art performance in semantic, in-context, and interactive segmentation.

**Strengths:**

1. Unified design integrating semantic, in-context, and interactive segmentation.
2. Consistent performance gain across diverse datasets and modalities.

**Weaknesses:**

The novelty of the paper is somewhat unclear. It appears to mainly combine existing ideas from interactive segmentation models (e.g., MedSAM) and in-context segmentation models (e.g., UniverSeg). It is not fully evident what new conceptual or methodological insight the proposed framework contributes beyond this integration. The authors should clarify the unique innovation or theoretical advancement that distinguishes K-Prism from prior works.

The three proposed “knowledge paradigms” — semantic priors, in-context examples, and interactive feedback — are somewhat loosely defined and may not convincingly qualify as distinct forms of knowledge. They are more simply like working mode instead of knoledge.The authors should better justify why these modes are conceptualized as “knowledge sources”.

The motivation for adopting a Mixture-of-Experts (MoE) design is insufficient. In medical segmentation, computational efficiency is typically not a critical bottleneck, so replacing a dense model with MoE seems unnecessary and possibly superficial.

**Questions:**

1. Authors should consider adding MedSAM for comparison
2. As shown in Table 2 and 3, the improvement is limited

---

> ### Author Response · Authors · 2025-11-20
> **Response to Reviewer 3nT9**
>
> We appreciate the reviewer’s comments and would like to clarify each of the concerns in detail:
> ## 1. Response to Weakness
> ### (1). Novelty and conceptual contribution
> First, we clarify that K-Prism’s contribution goes far beyond a simple combination of existing ideas. A core strength of our unified design is that the three knowledge paradigms—semantic, in-context, and interactive—produce synergistic benefits when integrated into a single framework. For example, using semantic (Mode-1) or in-context (Mode-2) predictions to initialize interactive refinement (Mode-3) reduces the number of clicks needed to reach 90% Dice by over 30% (Table 5(d)). In clinical practice, clinicians can start from an automatic mask and make quick corrections using our model; thus, there are fewer required interactions which can directly reduce workload and improve annotation efficiency. This synergy has not been explored in prior work, and to our knowledge, K-Prism is the first framework to achieve to learn the cooperative semantic, in-context, and interactive segmentation in a clinically meaningful way.
>
> Second, the Mixture-of-Experts (MoE) decoder is crucial for unified learning and multi-functional inference. As shown in Table 7(a), joint training across all modes achieves performance comparable to single-mode training while being three times more efficient. The MoE achieves this by routing semantic priors, exemplar cues, and click features—each residing in distinct feature spaces—through specialized expert pathways, preventing cross-mode interference and enabling dynamic, task-aware routing. This allows a single model to robustly support all three paradigms without sacrificing stability or accuracy.
>
> Third, prior approaches do not offer a strong basis for developing a unified and synergistic segmentation framework. For example, UniverSeg employs a lightweight interaction block that fuses reference and query features, which results in suboptimal in-context performance. MedSAM doesn’t introduce architectural innovation beyond SAM. It is finetuned on large-scale medical datasets and only supports bounding box prompts. In contrast, K-Prism proposes a principled dual-prompt formulation that integrates 1D sparse prompts representing task intent with 2D dense prompts providing spatial guidance, together with an explicit similarity-based conditioning mechanism that greatly improves few-shot adaptability. K-Prism further incorporates multiple essential components such as dual-prompt fusion, multi-scale feature conditioning, an attention-guided click–query mechanism, and an MoE decoder designed to avoid cross-mode interference. These elements collectively address the core challenges of unifying semantic, in-context, and interactive segmentation in a way that previous models have not achieved.
>
> We hope these clarifications adequately explain our motivation and innovations and fully address the reviewer’s concerns on novelty and contribution.
>
> ### (2). Characterization of knowledge paradigms
> We would like to clarify that our use of the term “knowledge paradigms” is not arbitrary. As discussed in the Introduction section, these three paradigms correspond to distinct and well-established knowledge sources in real clinical practice on medical image segmentation. Clinicians routinely rely on (1) semantic prior knowledge accumulated from years of exposure to labeled anatomy, (2) case-specific in-context knowledge obtained by reviewing similar historical cases when encountering rare or ambiguous findings, and (3) interactive feedback knowledge derived from iterative adjustments on a slice viewer during boundary refinement. These are genuinely different forms of knowledge that guide human expert decision-making in clinical practice. Importantly, these three knowledge sources map naturally and directly onto the three functional capabilities of our model, i.e., semantic segmentation, few-shot in-context segmentation, and interactive refinement. This conceptual correspondence is precisely why we adopt the term “knowledge paradigms”: it reflects the clinical practice origins of these cues and clarifies our motivation for designing a unified architecture that can represent and fuse them effectively.

---

> ### Author Response · Authors · 2025-11-20
> **Response to Reviewer 3nT9**
>
> ### (3). Regarding the motivation for the Mixture-of-Experts (MoE) design
> We appreciate the reviewer’s observation and agree that in most medical image segmentation workflows, computational efficiency is not the primary bottleneck, rather, achieving superior performance is the top priority. In fact, our motivation for adopting MoE is fully aligned with this perspective: the MoE in K-Prism is not introduced for speed, nor intended as a lightweight alternative to a dense decoder. Instead, the MoE is introduced to address a key challenge in unifying three very different segmentation paradigms: cross-mode feature interference. Semantic priors, in-context exemplars, and interactive clicks occupy different feature spaces, and forcing them through a single dense decoder leads to representation conflicts and degraded accuracy. The MoE mitigates this issue by routing each type of knowledge through specialized experts, allowing the model to allocate capacity according to the input source. This yields more stable and accurate performance across all modes without the need for separate architectures.
>
> To avoid misunderstanding, we emphasize that this design is fully consistent with the reviewer’s reasoning: (1) MoE is not used as a computational shortcut. (2) MoE is introduced because it is functionally necessary, not to optimize speed. (3) While MoE increases moderate model capacity, it yields significantly better performance, precisely matching the reviewer’s perspective that computing is not the main constraint in medical image segmentation.
>
> We hope this clarification shows that the point raised as a potential weakness is, in fact, a key strength of our approach. We respectfully believe this addresses the reviewer’s concern and clarifies the motivation behind our design.
>
> ## 2. Response to Question1
>
> We thank the reviewer for the constructive suggestion and acknowledge that MedSAM is an influential foundation model for medical image segmentation. Our original decision not to include MedSAM as a baseline was based on the two following considerations: (1) Training data overlap: MedSAM is pretrained on a large collection of medical datasets, many of which overlap with our evaluation benchmarks, making direct comparison with released weights potentially unfair; (2) Prompt mismatch: MedSAM is designed for bounding-box interaction, whereas our framework relies on sparse point prompts.
>
> Nevertheless, we appreciate the reviewer’s insight and have therefore included MedSAM in our experiments by retraining a point-prompt variant under the same interaction setting. Specifically, we initialized its ViT-Base encoder with the official MAE-pretrained weights and trained the entire model on our curated medical datasets using identical training configurations for a fair comparison. As shown in the revised manuscript, MedSAM achieves reasonable performance; however, its lack of a dense 2D prompt representation fundamentally limits its refinement capability, which is a limitation also reported in the SegNext paper when comparing SAM-style models against dense-prompt methods. In contrast, K-Prism consistently delivers the strongest results across all three segmentation paradigms. We have added the new results and corresponding discussion in the updated manuscript.
>
> We hope this clarifies our choice of baselines and addresses the concern regarding the comparison of MedSAM with our model.

---

> ### Author Response · Authors · 2025-11-20
>
> ## 3. Response to Question2
> We thank the reviewer for the comment. We would like to note that our experimental results actually show meaningful and consistent gains over prior methods across all three paradigms:
>
> Although semantic segmentation is not the primary focus of our method, K-Prism still achieves the best average Dice across 12 in-distribution datasets (86.21%) and obtains SOTA performance on external datasets (83.45%). On clinically challenging tumor segmentation tasks, the improvements remain substantial. In in-context segmentation, K-Prism achieves the highest average Dice (84.82%), outperforming Iris by +3.06% on average (84.82% vs. 81.76%). On several datasets, the relative improvements are even larger, e.g., on the unseen-class M&Ms-2 dataset, the improvement from 26.30% (Iris) to 41.61% (ours) corresponds to a relative gain of more than +15%.
>
> More importantly, as a multi-functional unified model, K-Prism is designed not merely for semantic or in-context segmentation, but to provide clinically useful interactive refinement, which is the mode clinicians rely on most when dealing with ambiguous boundaries or unsatisfied initial segmentation. In interactive segmentation mode, K-Prism provides the largest gains. Our model achieves (1) best Dice(5) across all settings, (2) highest click efficiency with lowest NoC90 / NoC95, and (3) fastest convergence among all baselines. These improvements are substantially larger than the semantic-only gains and highlight the effectiveness of our unified 1-D and 2-D prompt integration design.
>
> We also note that on mature benchmarks where strong baselines already achieve high Dice scores, recent methods typically improve performance by only a few points. In the semantic segmentation setting (Table 2), for example, the average Dice of Clip-driven, UniSeg, and Hermes lies in a narrow range (83.96–85.02%), and K-Prism further increases this to 86.21%. This margin is comparable to the improvements reported by Hermes, Iris, and other recent methods over their predecessors. Within this context, K-Prism’s advances represent meaningful progress, especially given that our unified framework achieves SOTA performance across semantic, in-context, and interactive segmentation modes, rather than being tailored to a single task.
>
> Given these results, we believe K-Prism’s advances in this paradigm represent meaningful and impactful contributions beyond limited improvement.
>
> ## 4. Summary
>
> We sincerely thank the reviewer for the detailed feedback and for highlighting important aspects of our work. Through the clarifications above, we hope that we have addressed the concerns regarding novelty, conceptual framing, MoE motivation, baseline selection, and performance improvements. We have provided additional experiments, new analyses, and further explanations that directly respond to each point raised. Collectively, these results reinforce that K-Prism is not a simple combination of existing components, but a unified and synergistic framework that delivers state-of-the-art performance across semantic, in-context, and interactive segmentation, with clear clinical relevance and methodological innovation.
>
> We hope the reviewer finds our clarifications satisfactory and agrees that the contributions of K-Prism are substantial. We kindly ask the reviewer to reconsider the evaluation in light of the additional evidence and explanations provided.

---

> > ### Comment · Reviewer_3nT9 · 2025-11-24
> >
> > Thank you very much to the authors for the clear clarifications. I have raised my score.
> >
> > Although I still do not fully agree with referring to the three types of clinical working modes as “knowledge” — since, from my perspective, knowledge is a more abstract and generalized concept rather than being strongly instance-linked, and the term “knowledge paradigms” feels somewhat overly packaged — I consider this only a minor conceptual inconsistency. My main concerns about the effectiveness of integrating the three types of segmentation prompting modes together have been addressed.
> >
> > Therefore, I agree that the paper can meet the ICLR publication criteria.

---

> > > ### Author Response · Authors · 2025-11-24
> > >
> > > Thank you very much for the positive reassessment and for raising the score. We sincerely appreciate your thoughtful engagement with our work and are glad that our clarifications have addressed your main concerns.
> > >
> > > Regarding the terminology of “knowledge” and “knowledge paradigms,” we appreciate your observation. We adopted this phrasing to mainly reflect the clinical conception underlying our design motivation, and we understand your concern regarding the strength of this conceptual wording. Thank you for highlighting this and we will carefully refine the terminology and conceptual framing in future revisions and works.
> > >
> > > We sincerely appreciate your constructive feedback and your recognition of the contribution of our work.

---

### Official Review · Reviewer_MRgi · 2025-11-01

**Soundness:** 4
**Presentation:** 4
**Contribution:** 3
**Rating:** 6
**Confidence:** 5

**Summary:**

The authors propose K-Prism, a unified segmentation framework capable of performing (1) standard segmentation on in-class labels, (2) in-context learning–based segmentation using reference exemplars, and (3) interactive segmentation through user feedback. This is achieved by learning a 1-D sparse prompt that encodes what to segment and a 2-D dense prompt that encodes where to attend, which are jointly processed through a Mixture-of-Experts decoder for dynamic, task-aware feature routing. Experiments conducted on 18 public datasets spanning CT, MRI, X-ray, pathology, and ultrasound demonstrate that K-Prism achieves state-of-the-art performance across all three segmentation paradigms, with generalization to unseen classes and external datasets.

**Strengths:**

1. K-Prism unifies semantic, in-context, and interactive segmentation within a single framework through a novel design that leverages 1-D and 2-D prompts to encode complementary information about what and where to segment.
2. The use of a Mixture-of-Experts decoder is well-motivated, enabling task-aware specialization and effective fusion of different prompt types while maintaining a shared representation space.
3. The paper is well-written and easy to follow, with clear methodological explanations and thorough experimental validation that make the work accessible and reproducible.

**Weaknesses:**

1. While K-Prism performs well on in-distribution and cross-dataset tests, its accuracy drops notably for unseen anatomical classes, suggesting limited generalization beyond trained label types.
2. The paper does not compare against nnU-Net trained under equivalent conditions—either as a single generalist model trained on all datasets or as individual task-specific models. Such comparisons are important to validate that the proposed unification truly offers an advantage over established supervised baselines.

**Questions:**

1. The model shows noticeably lower accuracy on unseen anatomical classes compared to in-distribution results. Could the authors provide an analysis explaining why this gap occurs?
2. To better contextualize the proposed framework, can the authors include experiments comparing K-Prism to nnU-Net trained (a) jointly on all datasets as a generalist model, and (b) separately on each dataset as task-specific specialists?
3. The paper mentions that all competing methods are trained on the same dataset, but it is unclear how interactive baselines (e.g., SAM2, MultiverSeg) were handled. Were these models trained or fine-tuned from scratch, or initialized from existing pretrained weights?

---

> ### Author Response · Authors · 2025-11-20
> **Response to Reviewer MRgi**
>
> We sincerely thank the reviewer for the positive and insightful comments on our work. We appreciate the recognition of **K-Prism** unified design, dual-prompt representation, and strong experimental results across diverse modalities and segmentation paradigms. Below, we address the reviewer’s specific questions and provide further clarifications.
>
> ## 1. Response to Weakness 1 and Question1:
>
> We appreciate the reviewer’s insightful question. The performance gap between in-distribution and unseen anatomical classes is expected and can be explained by several factors as follows:
>
> ### (1). Interactive segmentation remains highly effective on unseen-class datasets.
>
> The performance drop on unseen classes is indeed noticeable, but it does not impact all modes equally. Our interactive mode remains highly robust, delivering strong results even on completely unseen anatomy and clearly outperforming all baselines (Table 4). For example, on the unseen-class datasets, K-Prism achieves 90.67\% Dice(5) and NoC90 = 4.32, the best among all compared methods, showing that the underlying representation remains highly adaptable. While the performance is lower than on in-distribution data, this drop is expected and reasonable. This robustness is expected because interactive segmentation does not rely solely on learned semantic priors. Each user click provides localized appearance cues, i.e., texture, boundaries, and intensity patterns, together with precise spatial constraints. With our dual-prompt design, K-Prism learns a stable mapping from click features to mask refinement, enabling rapid and accurate corrections even under extreme out-of-distribution clinical scenarios.
>
> ### (2). In-context segmentation on completely unseen domains is inherently difficult.
> The performance drops are more pronounced in the in-context mode. However, in-context segmentation on fully unseen anatomical domains is an intrinsically challenging problem. All in-context baselines show degraded and unstable performance on unseen-class datasets in our experiments. In in-context mode, the model must infer both a 1-D abstract object summary ("what to segment") and a 2-D spatial query-reference matching/alignment (``where to segment'') from a single exemplar and mismatching between query and reference is the main reason of the performance drop. When the target anatomy has never appeared during training, the 1-D summary may become semantically irrelevant, and the 2-D alignment becomes highly ambiguous. Other latest in-context methods struggle under this setting: for example, Iris only achieves $<$ 30\% Dice on unseen pancreas segmentation, and UniverSeg exhibits similarly low performance, even on some external datasets that are still within the same broad domain. These results reflect the inherent difficulty of exemplar-based in-context segmentation under severe anatomical and distribution shifts.
>
> ### (3). The difficulty varies substantially across datasets.
>
> BraTS (brain MRI) introduces an extreme domain shift, as our training corpus contains no brain CT or MRI, causing all in-context methods to show degraded and unstable performance. In contrast, M\&Ms-2 (unseen long-axis cardiac MRI) is unseen but still shares partial distribution overlap with short-axis cardiac MRI datasets seen during training (e.g., M\&Ms). In this setting, K-Prism achieves the strongest performance among all baselines, demonstrating that even limited overlap in anatomy or modality is sufficient for our architecture to generalize effectively across domains.
>
> Lastly, and importantly, the unified design of in-context learning and interactive refinement equips our model the potential for improved performance under completely unseen scenario. If users are unsatisfied with the in-context segmentation results, fast interactive refinement (see discussion in the previous perspective) can be performed using Mode-3 to achieve clinical demands. We highlight this unique clinically practical design of our unified model in contrast to fragmental models.
>
> We will incorporate the above analysis into the revised manuscript, and we hope it satisfactorily addresses the concerns regarding unseen-class behavior of our model.

---

> ### Author Response · Authors · 2025-11-20
> **Response to Reviewer MRgi**
>
> ## 2. Response to Weakness 2 and Question 2
> We thank the reviewer for raising this important and highly relevant point. nnU-Net is indeed a widely recognized benchmark in medical image segmentation. However, nnU-Net is not designed for universal joint training across heterogeneous datasets. This is also why several recent works, such as UniSeg, introduce additional architectural modules to extend nnU-Net into a multi-dataset or universal setting.
>
> Given this, we believe that the most appropriate comparison is the specialist nnU-Net setting, where a separate nnU-Net model is trained on each dataset. We have therefore begun training specialist nnU-Nets for all datasets using the same preprocessing, augmentation, and compute budget as K-Prism. Since this requires training many independent models, the full results are still in progress. We will include the complete comparison in the revised manuscript.
>
>
> ## 3. Response to Question 3
>
> We thank the reviewer for raising this insightful point. For SAM2, which was pretrained on large-scale natural images, we initialized the model using the official *sam2.1_hiera_base_plus* checkpoint and fine-tuned it on our curated medical datasets under the same training settings as all other methods. For SegNext, the encoder was initialized with *MAE-pretrained ViT-base weights* and then jointly trained with the decoder on our medical image datasets using the same setting. For other medical segmentation models such as MultiverSeg, to avoid any potential data overlap with their original training sources, we trained them from scratch on our curated datasets to ensure full convergence. User interactions were simulated consistently across all methods. Additional training details have been provided in Appendix A.2 for clarity.
>
> ## 4. Summary
>
> We sincerely thank the reviewer for the thoughtful and detailed feedback. Across the three questions, we have provided new analyses, additional experiments, and clarifications to resolve the reviewer’s concerns. In particular:
> (1) we explained the observed performance gap on unseen anatomical classes with mode-specific reasoning and demonstrated the robustness of our interactive mode;
> (2) we clarified the role of nnU-Net and have initiated specialist nnU-Net experiments following the reviewer’s suggestion; and
> (3) we detailed the training and initialization strategies for all interactive baselines, including SAM2, SegNext, and MultiverSeg.
>
> We hope that these expanded analyses and updates strengthen the manuscript and fully address the reviewer’s concerns. We greatly appreciate the reviewer’s constructive comments, which have helped us further improve the clarity and completeness of this work.

---

> > ### Author Response · Authors · 2025-11-27
> >
> > ## Follow-up to Question 2
> >
> > We thank the reviewer for the thoughtful suggestion regarding the inclusion of nnU-Net as a task-specific baseline. We have completed the additional nnU-Net experiments and integrated the results into the revised manuscript. The corresponding analysis has been added to Section 4.2.1 (Semantic Segmentation), where we compare task-specific and universal models and discuss their respective strengths and limitations. We sincerely appreciate the reviewer’s patience and valuable feedback, which helped strengthen the completeness and clarity of our work.

---

### Official Review · Reviewer_ZEBx · 2025-11-01

**Soundness:** 3
**Presentation:** 3
**Contribution:** 3
**Rating:** 6
**Confidence:** 2

**Summary:**

This paper presents K-Prism, a unified framework for medical image segmentation that systematically integrates three clinically relevant knowledge paradigms: (i) semantic priors learned from annotated datasets, (ii) in-context knowledge from few-shot reference examples, and (iii) interactive feedback from user inputs.
The key design lies in a dual-prompt representation—1-D sparse prompts encoding “what to segment” and 2-D dense prompts encoding “where to attend”—together with a Mixture-of-Experts (MoE) decoder for dynamic, task-aware routing.
K-Prism supports three operational modes (semantic, in-context, interactive) within a single architecture and achieves state-of-the-art results across 18 datasets spanning CT, MRI, X-ray, ultrasound, pathology, and fundus imaging. The model demonstrates strong generalization to external and unseen-class datasets and enables efficient refinement through user interaction.

**Strengths:**

- The paper addresses a clear and practical gap: the fragmentation of current medical segmentation models across different knowledge paradigms.
- The dual-prompt + MoE design is conceptually elegant and technically sound, enabling unified training and inference across tasks without architectural modifications.
- Experiments are comprehensive and convincing, covering 18 public datasets, multiple modalities, and both in-distribution and cross-dataset evaluations.
- K-Prism achieves consistent SOTA performance across semantic, few-shot, and interactive settings, with particularly impressive efficiency in interactive segmentation (lowest NoC90 and NoC95).
- Ablation and analysis are thorough, confirming the contribution of each component and the dynamic specialization of experts.
- The work has practical clinical potential, as it can streamline deployment and annotation pipelines.

**Weaknesses:**

- All experiments are conducted on 2D slices, while many clinical workflows require 3D volumetric segmentation.
- The method relies on the quality and availability of reference examples for in-context mode, which may constrain performance in data-scarce or noisy scenarios.
- Discussion on model interpretability or failure cases is relatively limited.

**Questions:**

- Could the framework be extended to 3D volumetric segmentation without major architectural changes?
- How significant is the computational cost of the MoE decoder compared to standard transformer-based decoders (e.g., in inference latency)?
- Is there any strategy to automatically balance or select the optimal knowledge source (semantic / in-context / interactive) during inference, especially when user feedback is unavailable?

---

> ### Author Response · Authors · 2025-11-20
> **Response to Reviewer ZEBx**
>
> We sincerely thank the reviewer for the positive and insightful comments on the strengths, clarity, and comprehensiveness of our work. We appreciate the recognition of K-Prism’s unified design, dual-prompt representation, and strong empirical results across diverse imaging modalities and segmentation paradigms. Below we address the reviewer’s specific concerns and provide further clarifications.
>
> ## 1. Response to Weakness 1 and Question 1
>
> We thank the reviewer for this clinically meaningful question.
>
> First, our framework can be naturally extended to 3D volumetric segmentation, as both the dual-prompt formulation and the MoE decoder are dimension-agnostic. Extending from 2D to 3D simply requires replacing 2D patch embedding and convolution layers with their 3D counterparts, without any structural or training changes.
>
> Second, our focus on 2D segmentation reflects real-world clinical workflows, as highlighted by insights from our clinical collaborators indicating that radiologists mainly perform boundary refinement on 2D slices, where fine-grained organ, lesion, or tumor boundaries can be precisely adjusted. Fully 3D interaction is less practical, as volumetric rendering can obscure internal structures. While semantic and in-context segmentation naturally generalize to 3D, interactive refinement remains clinically most meaningful in 2D, motivating our choice of slice-based design and experiments.
>
> Finally, we plan to explore a full 3D extension in future work. Specifically, we aim to investigate efficient 2D-to-3D propagation strategies, which use accurate 2D interactive refinement to update the entire 3D volume. A concise version of this discussion has been incorporated into the revised Discussion section.
>
> ## 2. Response to Weakness 2
>
> We appreciate the reviewer’s sharp insight on this point. To assess how sensitive Mode-2 (in-context) segmentation is to the quality of the reference exemplar, we conducted an additional experiment on the ACDC and BUS datasets.
>
> For each dataset, we selected 10 distinct reference exemplars (independent images/cases). Using each exemplar as the sole reference, we performed a full 1-shot in-context inference on all query images, yielding 10 independent evaluations per dataset. We then computed the **mean, standard deviation, minimum, and maximum** Dice scores to quantify performance sensitivity.
>
> As shown in the table below, the variance across reference exemplars is very small for BUS (78.43 ± 0.72), indicating that the model behaves very stably in 2D settings. For ACDC (83.68 ± 2.38), the variance is slightly larger due to more complex 3D anatomical variability, but remains moderate and within an acceptable margin. Importantly, our main results demonstrate that even under strict 1-shot conditions—where reference quality and data scarcity are known challenges—K-Prism maintains strong and reliable in-context performance.
>
> | Dataset | Mean | Std | Min | Max |
> |--------|------|------|------|------|
> | **BUS**  | 78.43 | 0.72 | 77.09 | 79.15 |
> | **ACDC** | 83.68 | 2.38 | 79.10 | 86.18 |
>
> We hope this analysis addresses the reviewer’s concern regarding Mode-2 sensitivity and supports the practical reliability of our in-context segmentation design. Detailed experimental setup and further analysis have been added to **Appendix A.4**.
>
> ## 3.Response to Question 2
>
> We thank the reviewer for highlighting the importance of inference efficiency in clinical workflows. To address this, we measured per-image throughput on the ACDC dataset using a single A100 GPU with varying numbers of MoE experts:
>
> | # of Experts | 5 | 4 | 3 | 2 | 1 (Plain Decoder) |
> |--------------|---|---|---|---|--------------------|
> | FPS          | 3.63 | 4.24 | 4.61 | 5.10 | 5.38 |
>
> These results demonstrate that the MoE decoder moderately reduces inference speed. However, it delivers substantial and consistent performance improvements, as evidenced in Table 7(b). In practical clinical deployment on multi-GPU servers, throughput would scale proportionally, and adding more experts would not compromise real-time usability. Detailed experimental setup and additional analysis are added in Appendix A.4.
>
> ## 4. Response to Weakness3
> We appreciate the reviewer’s comment regarding the need for deeper discussion on model behavior and failure cases. In response, we conducted an additional analysis for unseen anatomical classes to better understand the observed performance gap. This analysis clarifies why such differences arise under anatomical and domain shifts and highlights that the interactive mode remains substantially more robust on unseen anatomy. We have incorporated this discussion into the revised manuscript to improve interpretability and provide clearer guidance on where the model performs reliably. In the updated Appendix, we also include representative visual examples and a brief failure-mode discussion to further illustrate typical error patterns and model behavior under challenging scenarios.

---

> ### Author Response · Authors · 2025-11-20
> **Response to Reviewer ZEBx**
>
> ## 5. Response to Question3
>
> This is an excellent and clinically relevant question. The choice of knowledge source at inference naturally depends on the clinical scenario. When user feedback is unavailable, the interactive mode is simply not applicable; if the target anatomy is unknown but reference exemplars are available, in-context segmentation is preferred; and if the class was seen during training, semantic segmentation is typically the most reliable choice. Importantly, K-Prism is designed to support all these working modes within a single unified model, allowing clinicians to select the most appropriate strategy based on their workflow rather than imposing a fixed operation mode. If full automation is desired, deployment can easily incorporate a lightweight script to detect available inputs (e.g., clicks, exemplars) and automatically route the model to the corresponding mode, making this a practical rather than fundamental limitation.

---

> ### Author Response · Authors · 2025-11-20
> **Response to Reviewer ZEBx**
>
> ## 6. Summary
> We sincerely thank the reviewer for the constructive and clinically meaningful feedback. We appreciate the positive assessment of our work and the opportunity to clarify several important aspects of K-Prism. In this rebuttal, we have addressed each concern in detail, including (i) the motivation behind our 2D design and its relationship to clinical workflows, (ii) the robustness of Mode-2 in-context segmentation with respect to reference exemplar quality, (iii) the computational overhead of the MoE decoder with concrete throughput measurements, and (iv) additional analysis and failure-case discussion explaining the behavior on unseen anatomical classes. We hope that these clarifications, together with the new experiments and extended discussions added to the revised manuscript, effectively resolve the reviewer’s concerns and further highlight the practicality, robustness, and clinical relevance of our unified K-Prism framework.

---

### Author Response · Authors · 2025-11-20
**First Overall Summary**

We sincerely thank all reviewers and the area chair for their detailed feedback and constructive suggestions. In this rebuttal, we have substantially strengthened our paper by:

(1) providing new analyses on unseen-class behavior, expert specialization, in-context exemplar sensitivity, extended discussion of 3D extensibility, and model interpretability, including detailed evaluations of how reference quality and anatomical/domain shifts affect Mode-2 performance;

(2) adding multiple new experiments, including robustness to exemplar selection, few-shot reference scaling, throughput and computational efficiency measurements under different MoE configurations, and additional comparisons with MedSAM and specialist nnU-Net models;

 (3) offering deeper explanation of our unified design, including why the dual-prompt formulation and MoE decoder naturally support 3D extension, how the architecture mitigates cross-mode interference, and the clinical relevance of integrating semantic, in-context, and interactive segmentation within one unified framework;

(4) revising the manuscript to improve clarity, expand discussion, and incorporate visual examples, new tables, and additional results in the appendix.

We appreciate the reviewers’ thoughtful comments, which have significantly improved the quality, completeness, and practical grounding of our work. We hope the provided evidence and clarifications satisfactorily address all concerns, and we thank the reviewers and AC again for their time and insightful evaluation.

---

### Comment · Area_Chair_LKej · 2025-11-25

Dear Reviewers:

We kindly encourage you to take a moment to review the authors’ rebuttals and submit your feedback. Your prompt feedback is important for ensuring a thorough review. Thank you for your contributions to ICLR 2026. If you have responded to the authors' rebuttal, please feel free to ignore this message.

Thanks,
AC

---

### Author Response · Authors · 2025-11-28
**Updated Summary to the Area Chair After the Score Reversion**

Dear Area Chair,

Thank you very much for taking over the review process and for your time and effort in evaluating our submission. Due to a recent issue on OpenReview that required reassignment and score rollback, we want to provide a brief and consolidated summary to ensure you have full visibility into the major revisions, newly added analyses, and reviewer reactions following our rebuttal. We sincerely appreciate your support, and we hope this summary clearly conveys the updated status of the submission and the improvements made.

## 1. Substantial Additions and Strengthened Evidence
Across all reviewers’ questions, we have added extensive new analyses, experiments, and explanations:

1. Unseen-class analysis explaining the mode-specific performance gap and demonstrating the strong robustness of the interactive mode.

2. Reference-exemplar sensitivity study showing that Mode-2 performance is stable and not overly dependent on exemplar quality.

3. Few-shot scaling experiments (1/3/5/9-shot) demonstrating that Mode-2 improves steadily with more references, resolving the multi-organ bottleneck concern.

4. Evaluations of inference throughput under varying expert numbers, providing clarity on computational overhead and real-time feasibility.

5. Clarified clinical motivation for the 2D interactive workflow and the dimension-agnostic nature of extending the framework to 3D.

6. Added visual examples, failure-mode analysis, and improved appendix tables and figures to enhance interpretability and presentation.

7. Added specialist nnU-Net baseline results following reviewer suggestions, now included in Sec. 4.2.1.

8. Added MedSAM baseline following reviewer suggestions, with new results and corresponding discussion included in Sec. 4.2.3.

We believe these updates make the work more solid and further improve the quality of our manuscript.

## 2. Reviewer Responses and Updated Scores

1. Reviewer 3nT9 explicitly raised their score from 2 to 6, stating that their main concerns had been addressed and that the paper meets ICLR publication criteria. And the reviewer had already provided positive and supportive responses before the system-wide issue of Openreview was reported.

2. The other reviewers also have offered positive and supportive comments in their initial reviews, reflecting a consistent and favorable overall assessment of the work.

## 3. Overall Assessment
Following extensive revisions, the paper now features:

1. A unified and technically novel framework: K-Prism (dual-prompt + MoE).

2. Comprehensive and strengthened experiments across 18 datasets, 3 modes, and multiple modalities.

3. Clear clinical motivation and practical relevance.

4. Detailed analyses addressing generalization, robustness, and efficiency.

5. Consistently positive feedback from the reviewers, including updated scores.

We believe the submission now represents a strong, well-motivated, and empirically validated contribution that aligns well with ICLR standards.

We sincerely appreciate your time and attention, as well as the valuable feedback provided by the reviewers throughout the rebuttal process.

Best regards,

The authors

---

### Meta-Review · Area_Chair_WhZb · 2026-01-04

**Summary:**

This paper proposes K-Prism, a comprehensive framework for medical image segmentation. Experiments show that this method achieves state-of-the-art performance on 18 public datasets covering multiple imaging modalities, including CT, MRI, X-ray, ultrasound, pathology, and fundus images. The authors provided detailed responses to the reviewers’ comments. They clarified the design rationale, added supplementary experiments, and evaluated the inference speed for different MoE decoder configurations. Reviewer 3nT9 explicitly stated that they would increase the score during rebuttal, and the other reviewers gave relatively high scores.

**Reviewer Concerns:**

Reviewer ZEBx focused on the adaptation from 2D to 3D data and the computational overhead of MoE decoder.
Reviewer MRgi was primarily concerned with performance drop on unseen anatomical classes, comparisons with nnU-Net, and detailed implementation.
Reviewer 3nT9 questioned the clarity of novelty, conceptual formulation, and the motivation.
Reviewer GsDa raised concerns about 2D-to-3D adaptation and the computational cost of the MoE decoder, and further pointed out issues related to the in-context bottleneck and the quality of reference exemplars.
During rebuttal, the authors added detailed method descriptions, theoretical and performance analyses, as well as extensive ablation and comparative experiments in the revised manuscript. I believe, most concerns have been effectively addressed.

**Reviewer Scores:**

I believe that Reviewer 3nT9 will increase their score, and the other reviewers will keep their scores.

---

### Decision · Program_Chairs · 2026-01-26

Accept (Poster)